# Single-nucleus transcriptional and chromatin accessibility analyses of maturing mouse Achilles tendon uncover the molecular landscape of tendon stem/progenitor cells

Hiroki Tsutsumi[1†], Tomoki Chiba[1†], Yuta Fujii[1,2], Takahide Matsushima[1], Tsuyoshi Kimura[3,4], Akinori Kanai[5], Akio Kishida[3], Yutaka Suzuki[5], Hiroshi Asahara[1,2]*

[1]Department of Systems BioMedicine, Graduate School of Medical and Dental Sciences, Institute of Science Tokyo, Tokyo, Japan; [2]Department of Molecular and Cellular Biology, The Scripps Research Institute, La Jolla, United States; [3]Laboratory for Biomaterials and Bioengineering, Institute of Integrated Research, Institute of Science Tokyo, Tokyo, Japan; [4]Department of Biomedical Engineering, Faculty of Life Science, Toyo University, Saitama, Japan; [5]Laboratory of Systems Genomics, Department of Computational Biology and Medical Sciences, Graduate School of Frontier Sciences, The University of Tokyo, Chiba, Japan

*For correspondence:
asahara@scripps.edu

[†]These authors contributed equally to this work

Competing interest: The authors declare that no competing interests exist.

## eLife Assessment

This study presents a **valuable** finding of novel markers that may potentially identify resident tendon stem/progenitor cells (TSPCs). The study also presents a comprehensive single-cell transcriptional dataset that will be of value to the field. The evidence supporting the identification of novel markers of a TSPC is **incomplete**, requiring clarification of current analyses and additional validation experiments to demonstrate that these markers are indeed specific and these cells are indeed TSPCs. This work will be of interest to biologists and engineers focused on tendons and ligaments.

**Abstract** Tendons and ligaments are crucial connective tissues linking bones and muscles, yet achieving full functional recovery after injury remains challenging. We investigated the characteristics of tendon stem/progenitor cells (TSPCs) by focusing on the declining tendon repair capacity with growth. Using single-cell RNA sequencing on Achilles tendon cells from 2- and 6-week-old mice, we identified *Cd55* and *Cd248* as novel surface antigen markers for TSPCs. Combining single-cell RNA sequencing with single-nucleus RNA and ATAC sequencing analyses revealed that *Cd55*- and *Cd248*-positive fractions in tendon tissue represent TSPCs, as confirmed by their expression of established TSPC markers, with this population decreasing at 6 weeks. We also identified candidate upstream transcription factors regulating these fractions. Functional analyses of isolated CD55/CD248-positive cells demonstrated high clonogenic potential and tendon differentiation capacity, forming functional tendon-like tissue in vitro. This study establishes CD55 and CD248 as novel TSPC surface antigens, potentially advancing tendon regenerative medicine and contributing to the development of new treatment strategies for tendon and ligament injuries.

## Introduction

Tendons are connective tissues involved in the contractile movement of muscle to bone. Tendons are rich in extracellular matrix components, such as type 1 collagen and proteoglycans, which both have elastic and viscous properties to withstand overload (*Zhang et al., 2019*; *Mendias et al., 2012*; *Gumucio et al., 2015*; *Andarawis-Puri et al., 2015*; *Asahara et al., 2017*; *Connizzo et al., 2013*).

Tendons have a hierarchical structure. Collagen fibrils aggregate to form fibers, which then bundle into fascicles. Each fascicle is surrounded by a thin connective tissue layer called the endotenon. Fascicles are further enclosed by the epitenon, and in some cases, an additional outer layer called the paratenon. The number of fascicles and the presence of a paratenon can vary among different tendons and species (*Jozsa et al., 1991*; *Walia and Huang, 2019*).

Tendon and ligament injuries potentially account for the majority of musculoskeletal disorders and can lead to arthritis and spondylitis (*Gracey et al., 2020*). The most popular treatment approaches are surgery and conservation (*Steinmann et al., 2020*). However, full functional recovery is often not achieved due to scarring and fibrosis in the injured area (*Guilak et al., 2014*; *Thomopoulos et al., 2015*). Furthermore, the risk of postoperative rupture is high (*Leong et al., 2020*; *Loiacono et al., 2019*). In this context, tendon regeneration focused on tendon stem/progenitor cells (TSPCs) is attracting attention (*Leong and Sun, 2016*). Tendons have few cellular components, including tendon cells (tenocytes) and TSPCs (*Tang et al., 2016*; *Bi et al., 2007*). Tenocytes are responsible for tendon homeostasis, while TSPCs self-renew and differentiate into tenocytes. Thus, investigating TSPC biology is important for understanding tendon regeneration and homeostasis.

Murine TSPCs were first reported in 2007 by *Bi et al., 2007*. They demonstrated that TSPCs possess self-renewal capacity, colony-forming ability, and multi-differentiation potential in vitro. Furthermore, murine TSPCs were reported to simultaneously express stem cell markers such as *Cd44* and *Stem cells antigen-1* (*Ly6a*), as well as tendon-related genes like *Scleraxis* (*Scx*) and Collagen type I alpha 1 chain (*Col1a1*), and were identified as a subset of tendon cells within the tendon fascicle.

Subsequent research on murine TSPC localization has yielded diverse perspectives. Harvey et al. reported that *Tppp3/Pdgfra*-positive cells in the epitenon are induced upon tendon injury and contribute to repair (*Harvey et al., 2019*). Additionally, Yin et al. and Tempfer et al. demonstrated the presence of TSPCs around blood vessels (*Yin et al., 2016*; *Tempfer et al., 2009*).

Markers used to characterize murine TSPCs include CD73, CD105, and CD90, which are known criteria for mesenchymal stem cells (MSCs)(*Nusspaumer et al., 2017*; *Lui, 2015*). Additionally, TSPCs have been reported to express *POU class 5 homeobox 1* (*Oct-4*), *Nanog*, *Nucleostemin*, stage-specific embryonic antigen-4 (SSEA-4), *Myc*, *SRY-box transcription factor* (*Sox*2), *Fucosyltransferase 4* (*Fut4*), and other genes (*Zhang et al., 2021*). Expression of CD146 and CD44 has been confirmed as well (*Ruzzini et al., 2014*). The *Tppp3/Pdgfra*-positive TSPCs reported by Harvey et al. highly express *Cd34*, which is generally considered to have low expression in MSCs (*Harvey et al., 2019*; *Tachibana et al., 2022*). *Cd34* is known to be highly expressed in mouse embryonic limb buds at E14.5 compared to E11.5 (*Havis et al., 2014*), suggesting that *Cd34*-positive cells might reflect progenitor cells that constitute the limb bud, including tendons. However, these markers are not specific to TSPCs; thus, a definitive method to distinguish TSPCs from mature tendons in vivo has not been established yet (*Chen et al., 2014*; *Still et al., 2021*; *Li et al., 2021*; *Li et al., 2019*; *Wang et al., 2008*; *Chen et al., 2012*; *Cho et al., 2018*; *Fang et al., 2022*).

Regarding tendon regenerative capacity, Howell et al. reported an interesting observation. Tendons in juvenile mice can regenerate functional tissue after injury, but this ability is lost in mature mice, resulting in scar tissue formation (*Howell et al., 2017*). This finding suggests the possibility of abundant TSPCs in juvenile mouse tendons.

Given this background, we hypothesized that evaluating tendon tissue heterogeneity at the single-cell level using juvenile mouse tendons would enable a more detailed characterization of TSPCs. This approach is expected to advance the identification and characterization of TSPCs, which has been challenging with conventional methods.

Therefore, we performed single-cell RNA sequencing (scRNA-seq) using cells collected from 2- and 6-week-old mouse Achilles tendons and investigated clusters that co-express known TSPC markers, such as *Tppp3*, *Pdgfra*, and *Ly6a*. Then, we analyzed the expression of surface antigens in these clusters and identified *Cd55* and *Cd248* as novel candidate surface antigens for murine TSPCs.

Furthermore, snATAC-seq and snRNA-seq were performed simultaneously using cells collected from Achilles tendons to evaluate the validity of *Cd55* and *Cd248* as surface antigens for TSPCs and to identify the landscape of transcription factors (TFs) involved in tendon maturation. We sorted mouse Achilles tendon cells based on CD55 and CD248 and confirmed their phenotypes, demonstrating high clonogenicity and highly efficient induction into tendon cells. These results suggested that CD55 and CD248 are novel surface antigens of murine TSPCs and may be useful for understanding the process of tendon maturation and for applications in cell therapy.

## Results

### Identification of novel surface antigens for TSPCs

In mice, the healing capacity for tendon injuries is high up to 2 weeks of age, but decreases as the musculoskeletal system matures at 6 weeks (*Somerville et al., 2004*; *Howell et al., 2017*). To investigate whether this phenomenon is due to changes in the cellular population of tendon tissue, including fluctuations in progenitor cells, we performed scRNA-seq on mouse Achilles tendons at these two time points.

Achilles tendons were harvested from mice at 2 and 6 weeks of age. Following collagenase digestion and dead cell removal, we employed massively parallel, droplet-enabled scRNA-seq analysis (10x Genomics Chromium). Data processing was conducted using the Cell Ranger pipeline (10x Genomics). We analyzed 10,314 cells from 2-week-old mice (median 3167 genes/cell and 41,611 mean reads/cell) and 6513 cells from 6-week-old mice (median 732 genes/cell and 60,386 mean reads/cell). After doublet removal using DoubletFinder, we merged the datasets and performed unbiased clustering using Seurat, identifying a total of 15 clusters (*Figure 1A*). The top differentially expressed genes (DEGs) for each cluster, based on log2 fold change and statistical significance, are summarized in *Figure 1B*.

We identified two clusters (0 and 11) with high expression of *Mkx* and *Scx*, TFs involved in tendon development and growth. Cluster 0 showed higher expression of ECM-related genes such as *Col1a1* and *Fmod* compared to cluster 11, suggesting these were tenocytes (TC). Cluster 11, with high *Periostin* (*Postn*) expression (*Jacobson et al., 2020*), was classified as myotendinous junction (MTJ). Other clusters were characterized as follows: cartilage (CA, high *SRY-box transcription factor 9* [*Sox9*] expression), lymphocytes (LC1, LC2, high *Lysozyme 2* [*Lyz2*] and *Complement component 1, q subcomponent, alpha polypeptide* [*C1qa*]), endothelial cells (EC, high *Platelet and endothelial cell adhesion molecule 1* [*Pecam1*]), red blood cells (RBC, high *Hemoglobin alpha, adult chain 1* [*Hba-a1*]), smooth muscle cells (SM, high *Myosin light chain 9* [*Myl9*]), proliferating cells (PC, high *Mki67* and *Stathmin 1* [*Stmn1*]), Schwann cells (SW, high *Myelin basic protein* [*Mbp*]), vascular endothelial cells (VEC, high *Multimerin 1* [*Mmrn1*]), and macrophages (MC, high *Protein tyrosine phosphatase receptor type C* [*Ptprc*]). In cluster 3, *Col1a1* and *Fmod* were expressed; however, *Gm42418* and *AY036118* were highly expressed. These long non-coding RNAs are related to RN45s and therefore cluster 3 represents ribosomal contamination (*Isola et al., 2024*). It is also speculated that the cartilage cluster (CA) reflects enthesis, with expression of *Scx*, as well as *Sox9* (*Zhang et al., 2023*). DEGs of each cluster were summarized in *Supplementary files 1–15*.

To identify progenitor populations within these clusters, we analyzed expression patterns of previously reported markers *Tppp3* and *Pdgfra* (*Takakura et al., 1997*; *Morikawa et al., 2009*; *Harvey et al., 2019*; *Tachibana et al., 2022*), along with the known stem/progenitor cell marker *Ly6a* (*Holmes and Stanford, 2007*; *Sung et al., 2008*; *Hittinger et al., 2013*; *Sidney et al., 2014*; *Fang et al., 2022*; *Figure 1—figure supplement 1B*). We identified subclusters within clusters 1 and 4 showing high expression of these genes, which we defined as SP1 and SP2. SP2 exhibited the highest expression of these genes, suggesting it had the strongest progenitor characteristics. It has been reported (*Harvey et al., 2019*) that the *Tppp3*-positive population is localized to peritenon, and SP clusters might reflect peritenon as well.

SP2 also showed strong expression of genes associated with early tendon development in mouse embryos, as identified by RNA-seq of mouse limb tendon cells at E14.5 (*Havis et al., 2014*) and the Eurexpress mouse embryo transcriptome atlas database (*Diez-Roux et al., 2011*; *Figure 1—figure supplement 2A*). Gene Ontology (GO) analysis revealed upregulation of tendon development-related pathways in SP2, including TGFβ production and collagen-containing extracellular matrix

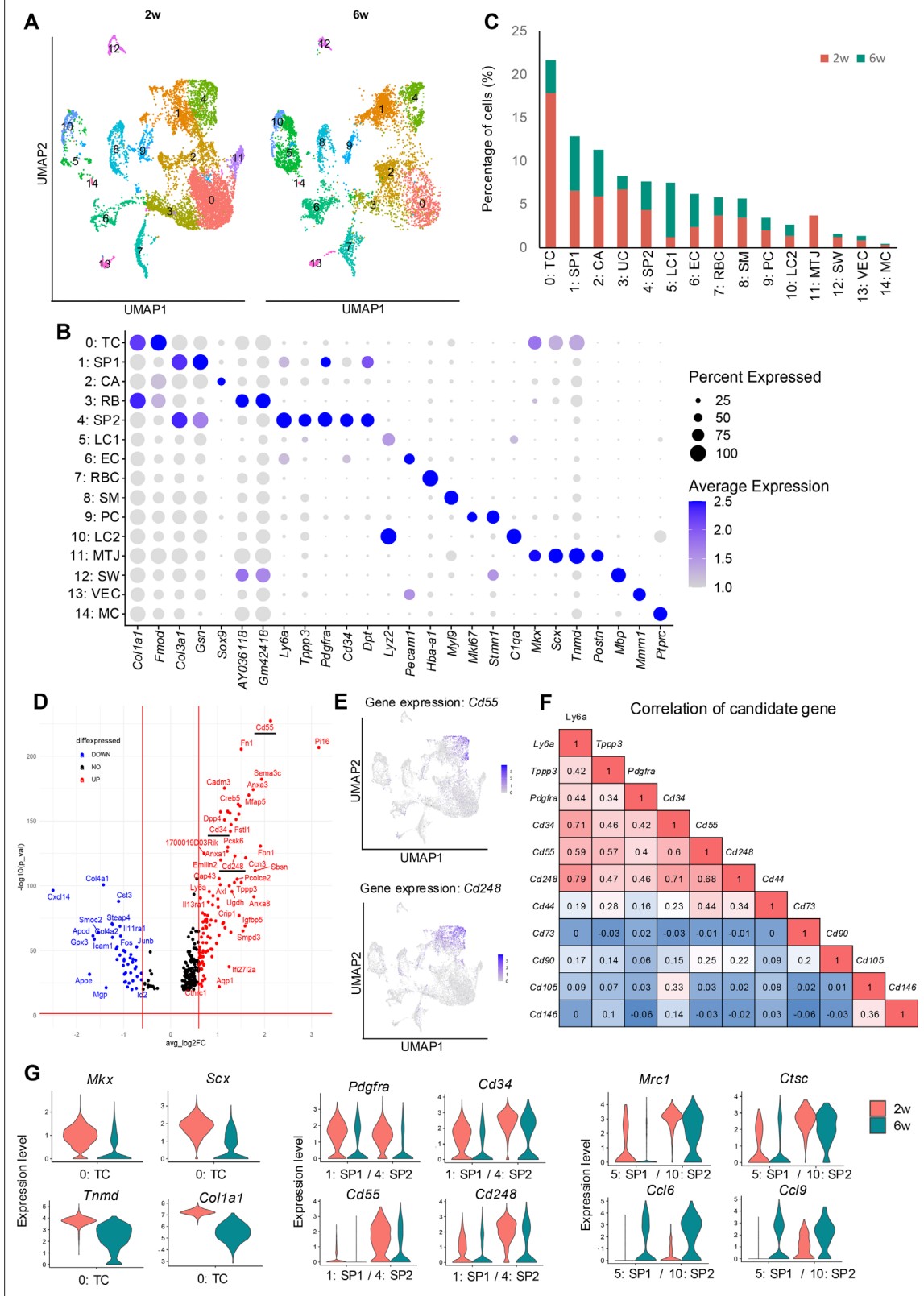

**Figure 1.** Single-cell RNA sequencing (scRNA-seq) of tendon cells from 2-week-old and 6-week-old mice and the identification of surface markers of tendon stem/progenitor cell (TSPC). (**A**) Integrated uniform manifold approximation and projection (UMAP) scRNA-seq clustering of cells harvested from 2-week-old and 6-week-old mouse Achilles tendons. (**B**) Dot plot of average gene expression levels of the indicated genes in each scRNA-seq cluster. The size of the dot reflects the percentage of cells in the cluster that express each gene. TC, tenocyte; SP1, tendon stem/progenitor cell_1; CA,

*Figure 1 continued on next page*

Figure 1 continued

cartilage; RB, ribosomal RNA; SP2, tendon stem/progenitor cell_2; LC1, lymphocyte_1; EC, endothelial cell; RBC, red blood cell; SM, smooth muscle cell; PC, proliferating cell; LC2, lymphocytes_2; MTJ, myotendinous junction cell; SW, Schwann cell; VEC, vascular endothelial cell; MC, macrophage. (**C**) Proportions of cells in clusters identified from scRNA-seq. Clusters are colored according to cluster type. (**D**) Volcano plot of gene expression in the SP2 cluster and the identification of candidate TSPC marker genes (red under line). (**E**) Feature plot of *Cd55* and *Cd248* expression. (**F**) Correlation of gene expression of TSPC candidate genes in 2-week data. (**G**) Violin plots presenting the gene expression changes for a selection of differentially expressed genes.

The online version of this article includes the following figure supplement(s) for figure 1:

**Figure supplement 1.** 2-Week and 6-week single-cell RNA sequencing (scRNA-seq) results.

**Figure supplement 2.** Comparison of 2-week and 6-week single-cell RNA sequencing (scRNA-seq) results.

**Figure supplement 3.** Comparative analysis of 2-week and 6-week single-cell RNA sequencing (scRNA-seq) results.

(*Figure 2—figure supplement 2B*). Evaluation of cell numbers in each cluster showed a decrease in SP2, TC, and MTJ at 6 weeks (*Figure 1C*, *Figure 1—figure supplement 1A*).

We then examined DEGs in SP2 compared to SP1 (*Figure 1D*). The whole list of DEGs comparing SP2 and SP1 was summarized in *Supplementary file 16*. Among these, we identified *Cd34*, *Cd55*, and *Cd248* as genes encoding surface antigens. *Cd34* is known to be highly expressed in mouse embryonic limb bud E14.5 compared to E11.5 (*Havis et al., 2014*), suggesting that *Cd34*-positive cluster might reflect a progenitor cell that consists of the limb bud, including tendons. However, *Cd55* and *Cd248* have not been discussed in this context. Expression of *Cd55* and *Cd248* was localized to areas corresponding to SP1 and SP2 (*Figure 1E*). We evaluated the correlation coefficients between the expression of these genes and previously suggested TSPC markers (*Cd73*, *Cd90*, *Cd105*, *Cd44*, *Cd146*), as well as *Tppp3*, *Pdgfra*, and *Ly6a* at 2 weeks of age. *Cd34*, *Cd55*, and *Cd248* showed high correlation, suggesting these genes as new candidate surface antigens for TSPCs (*Figure 1F*).

Comparing gene expression between 2 and 6 weeks, we observed a decrease in *Mkx* and *Scx* expression, consistent with previous reports (*Grinstein et al., 2019*). Furthermore, the expression of our candidate tendon progenitor cell markers *Cd34*, *Cd55*, and *Cd248* all decreased at 6 weeks. Inflammatory cells, including macrophages, increased at 6 weeks (*Figure 1G*). Gene expression analysis revealed a shift from M2 macrophage-related genes (*Mannose receptor C-type 1* [*Mrc1*] and *Cathepsin C* [*Ctsc*]) at 2 weeks to M1 macrophage-related genes (*Chemokine [C-C motif] ligand 6* [*Ccl6*] and *Chemokine (C-C motif) ligand 9* [*Ccl9*]) at 6 weeks. Comparative analysis between 2-week and 6-week tendons was summarized as a heatmap in *Figure 1—figure supplement 3*.

Analysis of cell-cell communication changes in postnatal tendon and surrounding tissues (*Lui and Wong, 2019*) using CellChat (*Jin et al., 2025*) showed a significant decrease in overall interaction at 6 weeks compared to 2 weeks. These results demonstrate that the cellular profile of tendon tissue changes dramatically at 6 weeks, with decreased expression of stem progenitor markers and reduced interaction with surrounding tissues.

## Single-nucleus RNA+ATAC analysis of 2-week-old mouse Achilles tendon cells

Gene regulation is mediated by the binding of TFs to cis-regulatory elements proximal to the gene. Consequently, epigenetic changes such as chromatin accessibility play a crucial role in gene expression (*Miller et al., 2014*). Moreover, TFs are typically expressed at low levels, potentially leading to false negatives in scRNA-seq due to detection limits. Chromatin accessibility changes often precede gene expression changes, potentially allowing us to predict transcriptional changes (*Ranzoni et al., 2021*). Therefore, we performed simultaneous ATAC-seq and RNA-seq at the single-cell level to identify gene expression changes and their associated epigenetic alterations, aiming to elucidate the postnatal growth mechanisms of tendon tissue and evaluate the validity of *Cd55* and *Cd248* as markers.

We extracted nuclei from cells isolated from 2-week-old mouse Achilles tendons and conducted droplet-enabled multi-omics analysis (10x Genomics Chromium), performing simultaneous snATAC-seq and snRNA-seq as a reference. We first analyzed the snATAC-seq data. After mapping reads to the genome and calling peaks, we annotated the location of peaks in terms of genomic features. The peaks were associated with promoters, introns, exons, and intergenic regions.

We evaluated 6571 cells (median high-quality fragments per cell: 17,408) and clustered them using latent semantic indexing and uniform manifold approximation and projection (UMAP) with the R package Signac (*Stuart et al., 2021*; *Figure 2A*). Given the limited knowledge of cell-specific chromatin accessibility, we assessed gene activity by computing counts per cell within the gene body and promoter of protein-coding genes. Using this gene activity data, we performed unsupervised clustering and identified 17 clusters (ground-truth annotation). Clusters expressing tendon-related genes such as *Mkx* and *Scx* were identified as clusters 1 and 6, defined as A1: TC1 and A6: TC2, respectively. Clusters 0, 14, and 15 were identified as expressing stem progenitor markers, including *Cd34*, *Ly6a*, *Pdgfra*, *Cd55*, and *Cd248*, and were defined as A2-0: SP1, A2-14: SP2, and A2-15: SP3, respectively. The characteristic gene activities and annotations for each cluster are summarized in *Figure 2B*.

Next, we analyzed the snRNA-seq data. We evaluated 10,314 cells (median 3167 genes/cell and 41,611 mean reads/cell). After doublet removal, we performed unbiased clustering using Seurat and identified 17 clusters (*Figure 2—figure supplement 1A*). Clusters expressing tendon-related genes *Mkx* and *Scx* were clusters 1 and 13, defined as R2-1: TC_1 and R2-13: TC_2, respectively. Clusters 2, 3, and 7 were identified as expressing stem/progenitor-related genes and defined as R2-2: SP_1, R2-3: SP_2, and R2-7: SP_3, respectively. To validate the consistency of our approaches, we conducted a comprehensive comparison between the cell clusters identified in our scRNA-seq analysis (which included both 2-week and 6-week samples) and our snRNA-seq analysis. We confirmed that all clusters identified in the scRNA-seq data could be similarly annotated in the snRNA-seq data, demonstrating strong concordance between these complementary approaches (*Figure 2—figure supplement 2*). This validation supports the reliability of our cell-type identification across different single-cell methodologies. Interestingly, R2-5: The MTJ cluster expressed not only *Tnc* but also *Sox9*, consistent with previous reports (*Nagakura et al., 2020*). Using the Signac package's 'FindTransferAnchors' function, we calculated predicted IDs for snATAC-seq clusters based on snRNA-seq annotations (predicted annotation) (*Figure 2C*). Evaluation of the two annotation methods for snATAC-seq revealed that both methods allowed annotation of major cell types, and correlation between the two was maintained. Thus, the validity of predicted annotations based on snRNA-seq was confirmed (*Figure 2D*).

R2-7: SP3, considered the most undifferentiated stem/progenitor fraction in snRNA-seq, could be divided into A2-0: SP1 and A2-14: SP2 in snATAC-seq.

Comparing gene activity of tendon-related and stem/progenitor genes in each cluster of the ground-truth annotation, A2-14: SP2 showed high expression of *Ly6a*, *Pdgfra*, and *Tppp3*, and inverse correlation with *Mkx*, *Scx*, and *Tnmd*. Furthermore, the newly identified surface antigens *Cd55* and *Cd248* also showed high expression in cluster 14. In contrast, *Cd34* did not show a high expression pattern in A2-14: SP2. These results suggest that A2-14: SP2 is the most immature cluster, with *Cd55* and *Cd248* showing characteristic expression (*Figure 2E*). Subcluster analysis of R7, considered the most immature fraction in snRNA-seq, revealed two clusters based on *Cd55*/*Cd248* and *Cd34* expression, similar to the snATAC-seq analysis. The former (R7-1) showed high expression of *Ly6a*, *Tppp3*, and *Pdgfra* (*Figure 2F*). Trajectory analysis using R2-7 and A2-14 as roots in snRNA-seq and snATAC-seq, respectively, showed increased expression of tendon-related genes *Mkx* and *Scx* along the trajectory, while expression of *Tppp3*, *Cd55*, and *Cd248* decreased (*Figure 3A*). Evaluation of peaks in each cluster based on gene expression levels in snRNA-seq revealed multiple peaks with heights correlated to gene expression for *Mkx* and *Scx*. A similar evaluation of *Cd55* and *Cd248* showed multiple peaks upstream of the transcription start site (TSS) for *Cd248* (*Figure 3B*). These results indicate correlation between snRNA-seq and snATAC-seq, and consistent with scRNA-seq results, suggest that A2-14 represents the most undifferentiated TSPCs based on *Cd55* and *Cd248* expression.

## Comparison of single-nucleus RNA+ATAC analysis between 2-week and 6-week mouse Achilles tendon cells

To confirm the presence of the identified *Cd55*/*Cd248* fraction at 6 weeks, we performed snRNA-seq and snATAC-seq on 6-week-old mouse Achilles tendon cells. Using the 2-week snATAC-seq data as a reference, we annotated the 6-week snATAC-seq clusters. Clusters A2-14 and A2-0 corresponded to clusters A6-12 and A6-0, respectively (*Figure 3—figure supplement 1*). We also annotated the 6-week snATAC-seq clusters using 2-week snRNA-seq with the same results (*Figure 3—figure supplement 2*). Gene activity evaluation revealed that, similar to the 2-week data, the 6-week A6-12 and A6-0 clusters showed *Cd55*/*Cd248* high and *Cd34* dim, and Cd55/*Cd248* dim and *Cd34* high

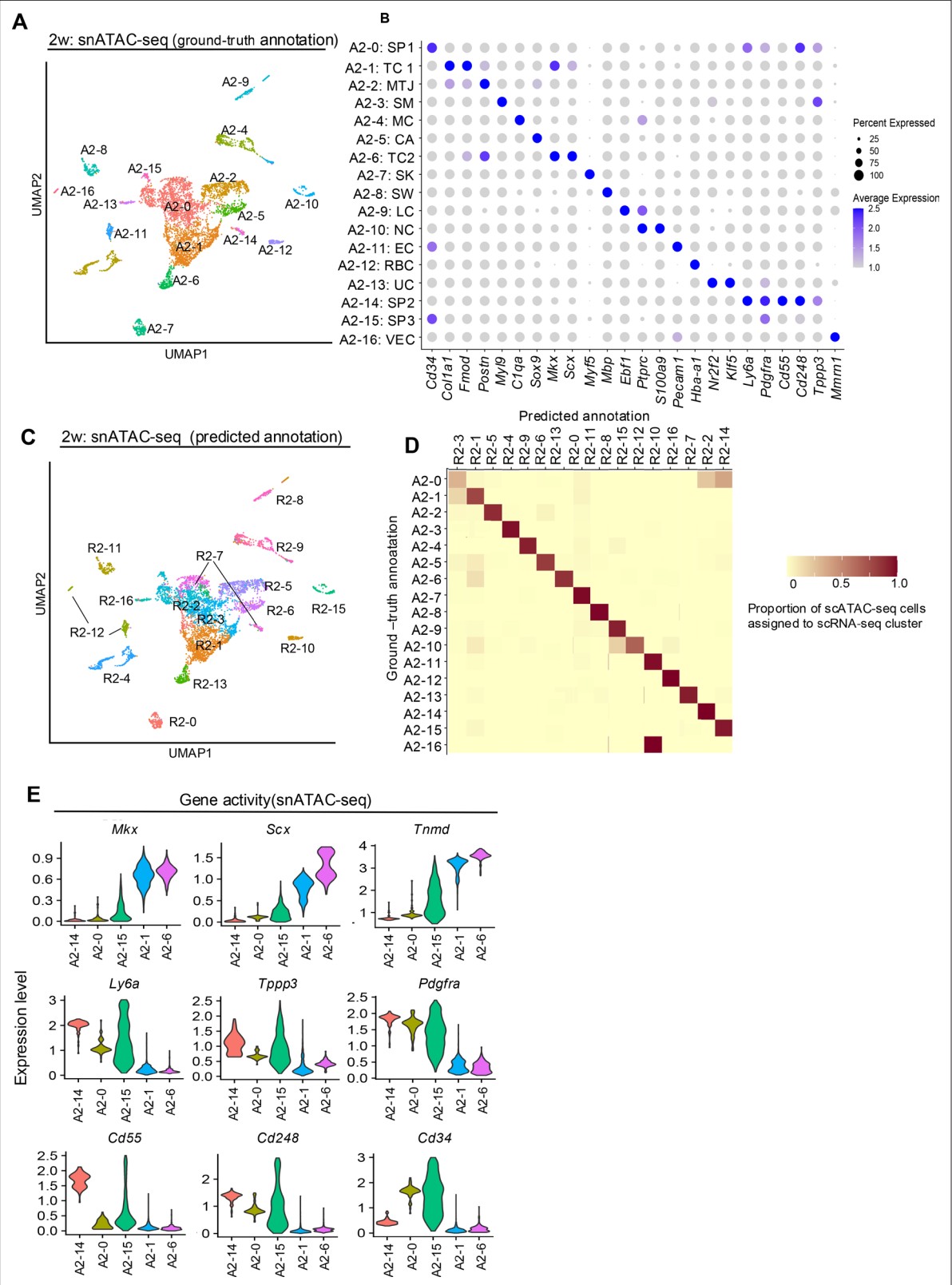

**Figure 2.** snATAC-seq of tendon cells from a 2-week-old mouse and the validation of Cd55 and Cd248 as candidate markers of tendon stem/progenitor cell (TSPC). (**A**) Uniform manifold approximation and projection (UMAP) snATAC-seq clustering of cells derived from the Achilles tendon of a 2-week-old mouse. Annotation was based on each gene activity (ground-truth annotation). SP1, tendon stem/progenitor cell_1; TC1, tenocyte_1; MTJ, myotendinous junction cell; SM, smooth muscle cell; MC, macrophage; CA, cartilage; TC2, tenocyte_2; SK, skeletal muscle cell; SW, Schwann cell; LC,

*Figure 2 continued*

lymphocyte; NC, neutrophil; EC, endothelial cell; RBC, red blood cell; UC, unspecified cell; SP2, tendon stem/progenitor cell_2; SP3, tendon stem/progenitor cell_3; VEC, vascular endothelial cell. (**B**) Dot plot of average gene activity of the indicated genes in each snATAC-seq cluster. The size of the dot reflects the percentage of cells in the cluster that express each gene. (**C**) UMAP visualization and predicted annotation of 2-week snATAC-seq after integration and label transfer of 2-week snRNA-seq data. (**D**) Identification of matching cell clusters between the 2-week snRNA- and 2-week snATAC-seq data from visualized as heatmap. The heatmap shows the proportions of cells from each snATAC-seq cluster across all sample conditions assigned to each snRNA-seq cluster as part of the label-transfer process. (**E**) Violin plot of tenocytes and TSPC-related gene expression in each cluster.

The online version of this article includes the following figure supplement(s) for figure 2:

**Figure supplement 1.** Analysis of 2-week snRNA-seq data.

**Figure supplement 2.** Analysis of 6-week snATAC-seq.

patterns, respectively. Furthermore, annotation of the 6-week snRNA-seq data using the 2-week snRNA-seq as a reference revealed clusters with expression patterns similar to those observed in snATAC-seq. In both cases, clusters with high *Cd55* and *Cd248* expression showed high gene activity (snATAC-seq) and gene expression (snRNA-seq) of *Tppp3*, *Pdgfra*, and *Ly6a*. These results confirm that the *Cd55* and *Cd248* high clusters identified in the 2-week snRNA-seq+snATAC-seq analysis are similarly detected at 6 weeks.

We also compared the snATAC-seq data between 2 and 6 weeks. No significant differences were observed in genomic annotations between the two time points (*Figure 3—figure supplement 3*). After merging the datasets (*Figure 3—figure supplement 3B*), we compared gene activity of tendon/stem-related genes to evaluate changes in gene activity. We found that the activity of *Tppp3*, *Pdgfra*, *Ly6a*, *Cd55*, *Cd248*, and *Cd34* all decreased at 6 weeks (*Figure 3—figure supplement 3C*). This was consistent with the decreased gene expression observed when comparing 2-week and 6-week data.

## Estimation of TF activity in 2-week mouse Achilles tendon cells

To estimate TF activity in each cluster, we used the Single-Cell Regulatory Network Inference and Clustering (SCENIC) package (*Aibar et al., 2017*) to calculate gene regulatory network activity from scRNA-seq gene expression data. SCENIC constructs gene expression networks centered on TFs and infers TF activity in each cluster. We summarized the predicted TF activities in tendon and stem/progenitor-related clusters R2-7, R2-2, R2-3, R2-1, and R2-13 identified by snRNA-seq (*Figure 4*).

Next, we evaluated the gene activity of these identified TFs in each snATAC-seq cluster, along with motif activity calculated by chromVAR (*Schep et al., 2017*). We also assessed the expression levels of TFs in each snRNA-seq cluster. In the most immature fractions, represented by cluster A2-14 in snATAC-seq and cluster R2-7 in snRNA-seq, *KLF transcription factor 3* (*Klf3*), *KLF transcription factor 4* (*Klf4*), and *cAMP responsive element binding protein 5* (*Creb5*) showed consistent behavior in gene activity, motif activity, and gene expression. Additionally, *Signal transducer and activator of transcription 2* (*Stat2*) and cAMP responsive element binding protein 3 like 1 (*Creb3l1*) showed high values in A2-0/A2-15 and A2-1, respectively. *Stat2* gene expression was not observed, likely due to false positives resulting from low expression levels. The gene activity and motif activity of each TF estimated by snATAC-seq correlated with the transcription factor activity calculated by SCENIC. Therefore, simultaneous analysis of snRNA-seq and snATAC-seq could be used to more accurately evaluate the function of TFs in each cluster.

Furthermore, SCENIC can predict candidate TFs regulating each gene (*Table 1*). *Cd55* and *Cd248* were predicted to be regulated by *Klf3* and *Klf4*, while *Mkx* and *Scx* were predicted to be under the control of *Creb3l1*. These predictions were consistent with the data calculated from gene activity and motif activity in snATAC-seq.

## In vitro evaluation of CD55 and CD248

Immunostaining evaluation of CD55 and CD248, the identified candidate stem/progenitor markers, revealed expression in the peritenon (*Figure 5C and A*). This was consistent with previous reports of localized expression of *Tppp3*/*Pdgfra*-positive cells in the peritenon and our analysis showing co-expression of *Tppp3*/*Pdgfra* and *Cd55*/*Cd248*. The results support the possibility that SP clusters reflect peritenon. Given that *Cd55* and *Cd248* expression appeared to reflect *Tppp3*, *Pdgfra*, and *Ly6a* expression more sensitively than *Cd34*, we extracted tendon cells from 2-week-old mice and sorted

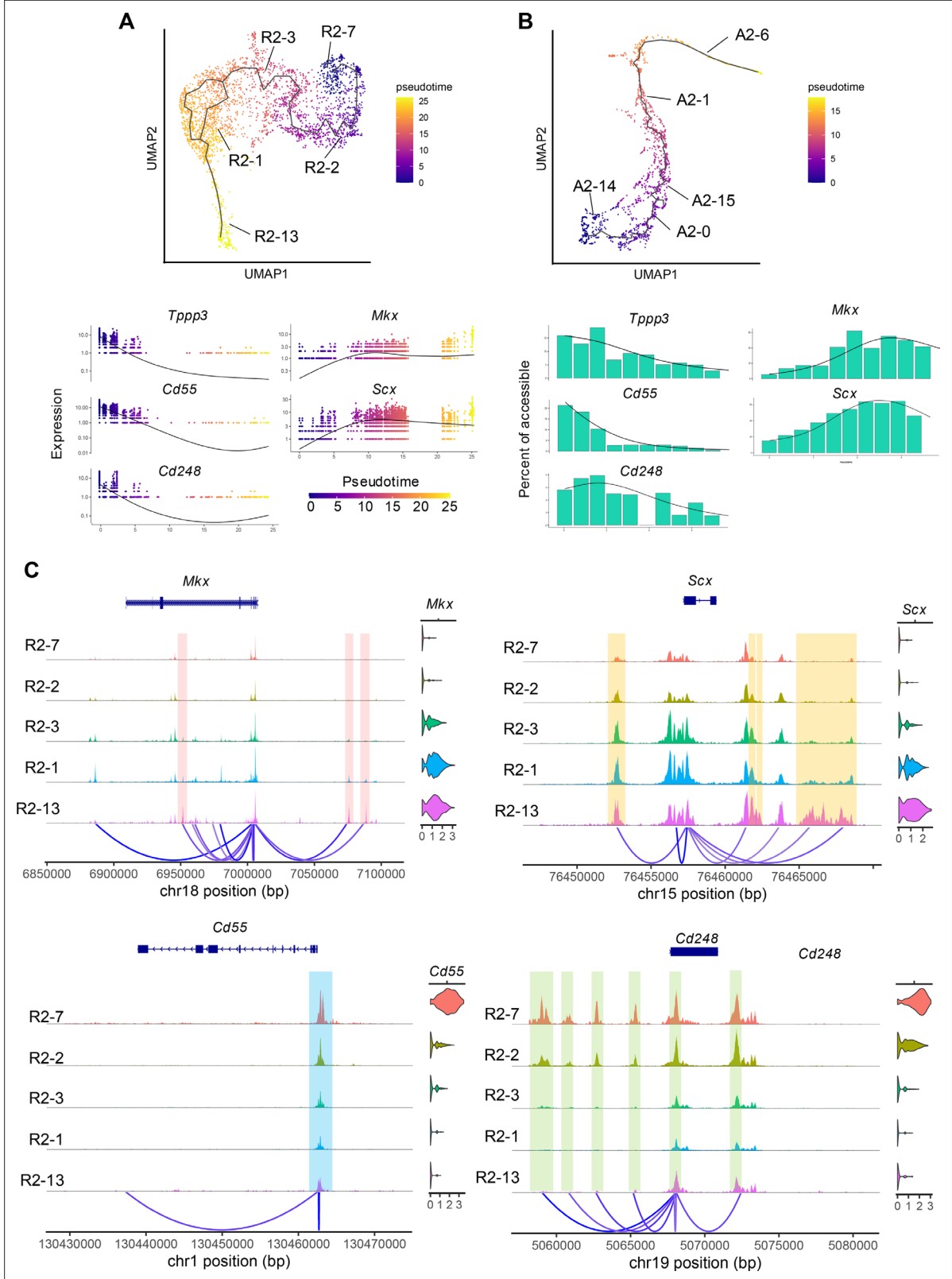

**Figure 3.** Trajectory analysis and peak visualization of snRNA-seq and snATAC-seq data for the tendon and tendon stem/progenitor cell (TSPC)-related cluster. (**A**) Uniform manifold approximation and projection (UMAP) representation of snRNA-seq differentiation trajectory of tenocytes and TSPC lineage and pseudotime-dependent gene expression changes of *Tppp3*, *Cd55*, *Cd248*, *Mkx*, and *Scx*, as inferred using Monocle3. (**B**) UMAP

*Figure 3 continued*

representation of snATAC-seq differentiation trajectory of tenocytes and the TSPC lineage and pseudotime-dependent gene expression changes, as inferred using Cicero. (**C**) Coverage plots of *Mkx*, *Scx*, *Cd55*, and *Cd248*. Selected peaks that differ across each cluster are highlighted.

The online version of this article includes the following figure supplement(s) for figure 3:

**Figure supplement 1.** Comparison of snRNA-seq and single-cell RNA sequencing (scRNA-seq).

**Figure supplement 2.** Analysis of 6-week snRNA-seq.

**Figure supplement 3.** Comparison of 2-week and 6-week snATAC-seq.

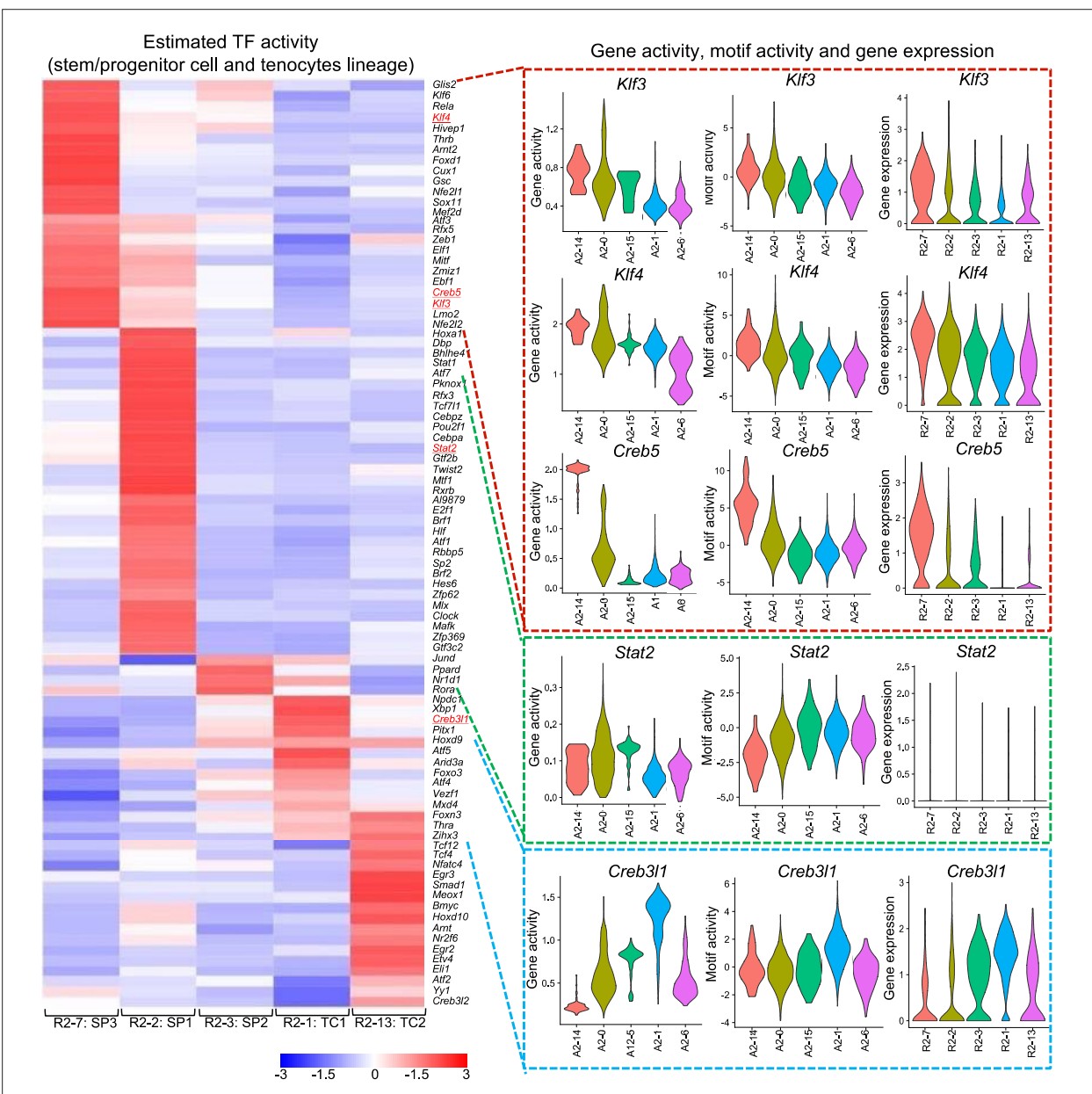

**Figure 4.** Transcription factor landscapes of 2-week mouse Achilles tendons. Single-Cell Regulatory Network Inference and Clustering (SCENIC) analysis of transcription factor activity based on 2-week snRNA-seq data for tenocytes and the tendon stem/progenitor cell (TSPC) lineage (left). Validation was performed based on the gene activity and motif activity of 2-week snATAC-seq and gene expression of 2-week snRNA-seq (right).

**Table 1.** Estimated transcription factor (TF) activity in each gene.

**Estimated TF activity: Cd55**

| TF | Spearman correlation |
| --- | --- |
| Klf4 | 0.264195541 |
| Klf3 | 0.262413411 |
| Zeb1 | 0.219593135 |
| Pbx1 | 0.204085218 |
| Hic1 | 0.202481054 |
| Klf2 | 0.196232716 |
| Nfe2l2 | 0.196076187 |

Estimated TF activity: Cd248

| TF | Spearman correlation |
| --- | --- |
| Hic1 | 0.375565336 |
| Klf3 | 0.277768834 |
| Zmiz1 | 0.272991765 |
| Klf4 | 0.260526766 |
| Zeb1 | 0.250583629 |
| Sp3 | 0.2502976 |
| Kdm5b | 0.193229132 |

Estimated TF activity: Mkx

| TF | Spearman correlation |
| --- | --- |
| Creb3l1 | 0.320638653 |
| Zfhx3 | 0.169341424 |
| Mxi1 | 0.149048856 |
| Bhlhe40 | 0.144301452 |
| Ets2 | 0.137445497 |
| Max | 0.116134957 |
| Elk3 | 0.098325126 |

Estimated TF activity: Scx

| TF | Spearman correlation |
| --- | --- |
| Creb3l1 | 0.316522613 |
| Npdc1 | 0.163452737 |
| Ets2 | 0.145463784 |
| Mix1 | 0.121248221 |
| Bhlhe41 | 0.115930755 |
| Elk3 | 0.10803805 |
| Bmyc | 0.104968457 |

them by fluorescence-activated cell sorting (FACS) to determine the biological phenotype of Cd55- and Cd248-positive cells (*Figure 5B*).

Gene expression analysis of sorted cells showed that CD55/CD248-positive cells, compared to -negative cells, had lower expression of *Mkx*, *Scx*, *Col1a1*, and *Creb3l1*, but higher expression of

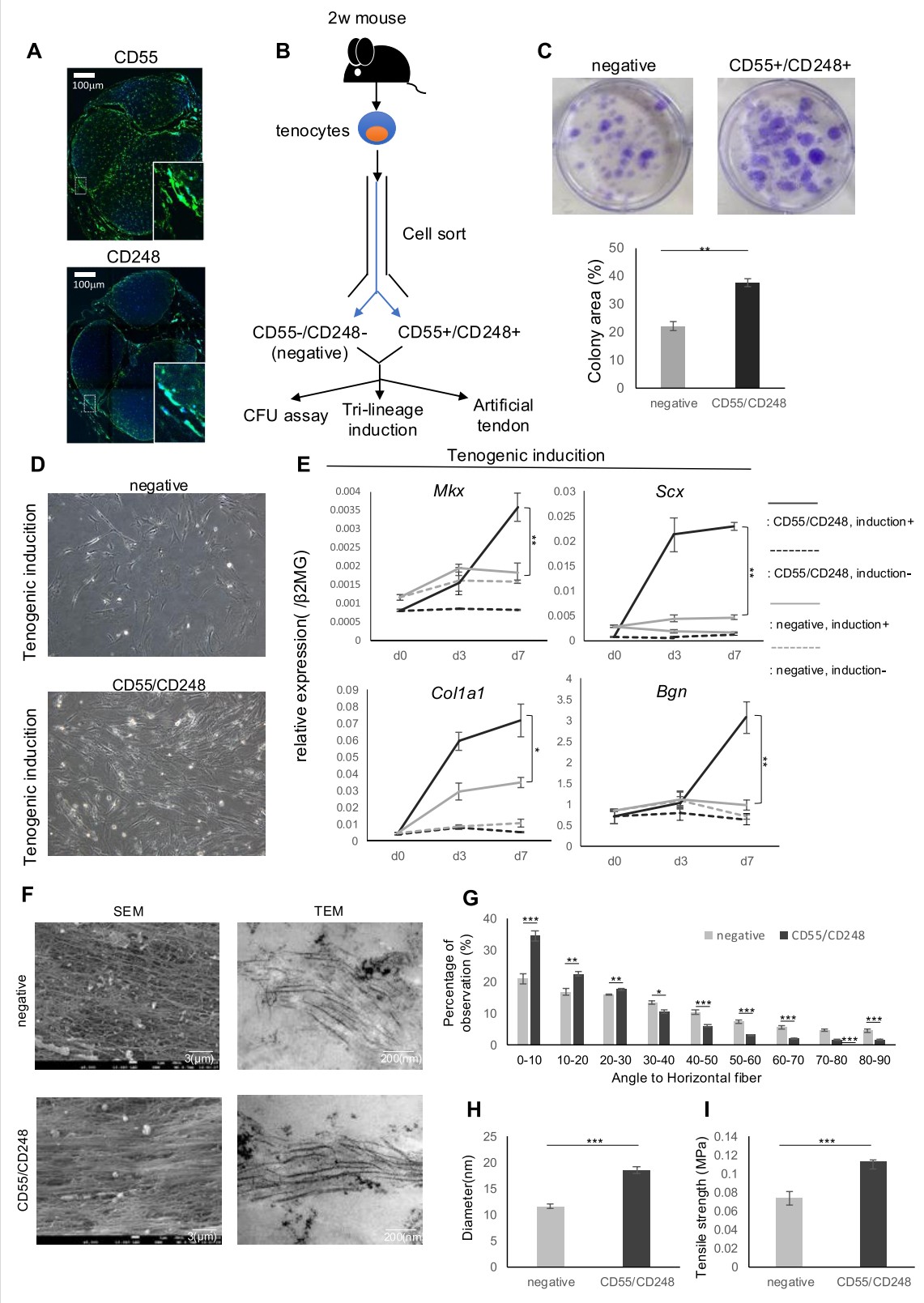

**Figure 5.** In vitro analysis of CD55+/CD248+ tendon stem/progenitor cells (TSPCs). (**A**) Immunohistochemical image of 10-week mouse Achilles tendons. Scale bars show 100 μm. CD55 and CD248, green; Hoechst 33342, blue. (**B**) Schema of the in vitro assessment of the capacity of CD55+/CD248+ TSPCs as the differentiation toward tenocytes. (**C**) Colony-forming efficiency of CD55+/CD248+ and CD55-/CD248- (negative) TSPCs. Colonies were stained with crystal violet (n=6). CD55+/CD248+ TSPC exhibited higher clonogenic capacity. Data are presented as means ± scanning electron

*Figure 5 continued on next page*

*Figure 5 continued*

microscopy (SEM). **p<0.01. (**D**) Morphological changes of CD55+/CD248+ and negative TSPCs after tenogenic induction. (**E**) Quantitative PCR of tendon-related gene expression in CD55+/CD248+ and negative TSPCs after tenogenic induction (n=3). Data are presented as means ± SD. **p<0.01, *p<0.05. (**F**) SEM and transmission electron microscopy (TEM) imaging of artificial tendons derived from CD55+/CD248+ and negative TSPCs. Data are presented as means ± SD. **p<0.01, *p<0.05. (**G**) Proportions of fiber alignment for each artificial tendon (n=4). Data are presented as means ± SEM. ***p<0.005, **p<0.01, *p<0.05. (**H**) Diameter of collagen fiber in each artificial tendon based on TEM imaging (n=4). Data are presented as means ± SD. ***p<0.005. (**I**) Tensile strength (MPa) of each artificial tendon (n=5). Data are presented as means ± SEM. ***p<0.005.

The online version of this article includes the following figure supplement(s) for figure 5:

**Figure supplement 1.** Gene expression changes in CD55+/CD248+ and negative tendon stem/progenitor cells (TSPCs).

**Figure supplement 2.** Chondrogenic and osteogenic induction of CD55+/CD248+ and negative tendon stem/progenitor cells (TSPCs).

*Ly6a*, *Tppp3*, *Pdgfra*, *Creb5*, and *Klf3*, consistent with snRNA-seq analysis (*Figure 5—figure supplement 1*). To assess stem/progenitor capacity, we performed colony formation assays. CD55/CD248 sorted cells showed significantly increased colony formation compared to CD55/CD248-negative cells (control) (*Figure 5C*).

We then evaluated the differentiation tendencies of these cells toward tendon, cartilage, and bone. Tendon differentiation resulted in clusters of spindle-shaped cells from CD55/CD248-positive cells, suggesting tenogenic differentiation (*Figure 5D*). Gene expression analysis revealed that while CD55/CD248-positive cells initially had lower *Mkx* and *Scx* expression than negative cells, this pattern reversed after differentiation (*Figure 5E*). Increased expression of other tendon-related genes such as *Col1a1* and *Bgn* was also observed. In contrast, no significant increases in *Sox9* and *Collagen type II alpha 1 chain* (*Col2a1*) (chondrogenic) *Craft et al., 2013* or *RUNX family transcription factor 2* (*Runx2*), and *Alkaline phosphatase, biomineralization associated* (*Alpl*) (osteogenic) expression were observed compared to negative cells during cartilage and bone differentiation assays (*Figure 5—figure supplement 2*). Cells negative for CD55/CD248 could be mixed cell populations, including hematopoietic lineages, cells from tendon mid-substance, immune cells, and/or endothelial cells. However, given that CD55/CD248-negative cells have increased expression of genes characteristic of tendons such as *Mkx*, *Scx*, and *Col1a1*, we could speculate that this popularity might reflect terminally differentiated tenocytes and CD55/CD248-positive cells possess tenogenic differentiation capacity.

To further evaluate the tenogenic potential of CD55/CD248-positive cells, we created tendon-like tissue (bio-tendon) using our previously reported 3D stretch stimulation culture system. Scanning electron microscopy (SEM) and transmission electron microscopy (TEM) analysis to assess collagen fiber density and thickness showed that bio-tendons derived from CD55/CD248-positive cells had an increased proportion of collagen fibers parallel to the stretch direction and increased collagen fiber diameter. TEM also revealed characteristic banding patterns and triple helix structures in these bio-tendons, indicating mature collagen organization (*Wieczorek et al., 2015*; *Figure 5F–H*). Stretch tests to evaluate the mechanical capacity of the bio-tendons showed high tensile strength (*Figure 5I*).

These results demonstrate that CD55/CD248-positive cells have a tendency to differentiate into tendon cells.

## Discussion

Tendons are known for their low cellular content, making complete functional recovery after injury challenging and increasing the risk of re-rupture. As a result, cell therapy has garnered attention as a novel treatment strategy, distinct from current conservative and surgical approaches. However, this requires a thorough understanding of TSPCs. Research on TSPCs has been limited due to the lack of identified characteristic surface antigens.

To our knowledge, no multi-omics analysis comparing juvenile and mature mouse tendon cells has been conducted. In this study, we performed single-cell RNA-seq along with snRNA-seq+snATAC-seq on nuclei isolated from 2-week and 6-week mouse Achilles tendons. While comparisons using older mice (e.g. 12 weeks or more) were initially considered, they were excluded due to extremely low cell yield and viability, making 2- and 6-week-old mice more suitable for this analysis. This approach allowed us to identify candidate surface antigens for TSPCs and elucidate the dynamism of TFs involved in post-developmental tendon growth. As a result, we discovered a novel combination of *Cd55* and *Cd248* as surface antigens, showing characteristic expression in the TSPC fraction.

While scRNA-seq analysis also showed *Cd34* expression characteristic of the TSPC fraction, consistent with previous reports, snRNA-seq+snATAC-seq analysis revealed that the *Cd34* high-expression cluster could be further classified into two clusters based on *Cd55* and *Cd248* expression patterns. The cluster showing high expression of both *Cd55* and *Cd248* also exhibited high expression of *Tppp3*, *Pdgfra*, and *Ly6a*, suggesting that isolating *Cd55*- and *Cd248*-positive fractions may more sensitively capture immature populations compared to *Cd34*.

snRNA-seq+snATAC-seq analysis provided simultaneous information on gene expression levels and open chromatin regions for each cluster. Notably, *Cd248* showed several peaks upstream of the TSS in high-expression clusters, likely reflecting its expression regulation mechanism. We also identified several peaks that increased proportionally with expression for *Mkx* and *Scx*. Given the many unknowns in *Mkx* and *Scx* expression regulation mechanisms (**Guerquin et al., 2013**; **Otabe et al., 2015**), we used the SCENIC package to predict upstream TFs based on expressed genes in each snRNA-seq cluster. Combined with gene activity and motif activity data from snATAC-seq, Creb3l1 was identified as a TF regulating *Mkx* and *Scx* expression. *Creb3l1* has been reported to increase in expression throughout development (**Liu et al., 2015**).

*Klf3*, *Klf4*, and *Creb5* were identified as characteristic TFs in the TSPC fraction. Recent reports have highlighted the role of the *Klf* family in limb development (**Kult et al., 2021**). Given that tendons are components of the developing limb, these TFs may have roles in tendon biology. *Creb5* has been reported to show increased expression from E11 to E13 (**Liu et al., 2015**), suggesting a role in early developmental stages. Furthermore, *Creb5* has been reported to regulate Proteoglycan 4 (*Prg4*) expression (**Zhang et al., 2021**). Further investigation into the functions of these TFs in TSPCs is necessary.

CD55 was identified by **Hoffmann, 1969** as a surface antigen functioning as a complement inhibitory factor on erythrocytes. CD55 is known to be expressed from early developmental stages and characterizes the initial differentiation stage of hematopoietic stem cells (**Guo et al., 2013**). It is also expressed in MSCs and has been reported as an early progenitor marker for mouse mammary epithelial cells (**Pal et al., 2017**). CD248 was identified in 1992 as an antigen for the FB5 antibody reacting with vascular wall cells (**Rettig et al., 1992**). CD248 is a transmembrane glycoprotein expressed in pericytes and fibroblasts during developmental stages. Regarding their relationship, CD55 and CD248 have been reported to be expressed in stromal cells during the early stages of arthritis (**Kendal et al., 2020**; **Choi et al., 2017**), but their detailed mechanisms in tendons remain unclear.

Previously suggested TSPC surface antigen candidates such as *Cd73*, *Cd90*, and *Cd105* showed poor correlation with the expression of genes like *Tppp3*. Moreover, the initial report on TSPCs (**Bi et al., 2007**) described TSPCs as *Scx+Cd34*-. The high expression of CD55 and CD248 in the peritenon, similar to *Tppp3* (**Harvey et al., 2019**; **Staverosky et al., 2009**), suggests the possibility of cells with tenogenic differentiation potential exhibiting *Scx+Cd34*- expression patterns within the tendon. Spatial information is important to investigate further. Future studies using lineage tracing experiments with mice labeled for CD55 and CD248 or mouse models of selective ablation of CD55 and CD248 are necessary to analyze the developmental functions of CD55- and CD248-positive cells and their roles in injury healing in more detail.

Clinically, to our knowledge, few studies have sorted TSPCs based on surface antigens and examined their in vitro tendon tissue-generating ability. In this study, we found that cells sorted for CD55/CD248 showed higher clonogenicity compared to CD55-/CD248- cells and demonstrated superior tendon tissue-generating ability in an artificial tendon model. In the future, CD55/CD248 double-positive TSPC cells may prove useful in clinical applications such as in vitro artificial tendon creation (**Cao et al., 2002**; **Tsutsumi et al., 2022**) and cell therapy for tendon injuries (**Huang et al., 2021**).

## Materials and methods
### Single-cell isolation
Mice were euthanized by exposure to $CO_2$ followed by cervical dislocation. To remove Achilles tendons, a longitudinal incision through the skin was made down the midline of the posterior aspect of the lower limb. A sharp transverse incision was made just distal to the myotendinous junction and again just proximal to the enthesis, and the Achilles tendon was carefully removed. The procedure was performed bilaterally. All animals were purchased from Sankyo Lab Service (Tokyo, Japan). Achilles

tendons were digested to obtain a single-cell suspension. In total, 40 or 60 (2 weeks and 6 weeks, respectively) mouse Achilles tendons were processed together as a single sample. After washing with 1× phosphate-buffered saline (PBS) several times and cutting 1-mm-wide sections using scissors, tendons were digested for 1 hr at 37°C with continuous shaking at 1200 rpm and 37°C in a dissociation solution consisting of 30 mg/mL collagenase (Wako, Osaka, Japan). After digestion, the single-cell suspension was filtered for debris using a 40 μm cell strainer and washed twice with 1× PBS. The samples were centrifuged for 15 min at 1500 rpm and 4°C. Dead cells were removed using a Dead Cell Removal Kit (Miltenyi Biotec, Bergisch Gladbach, Germany). Cell viability was 76% (2 week) and 73% (6 week), respectively. The experiment involving mice was approved by the Animal Experimental Committee of the Institute of Science Tokyo.

## scRNA-seq library construction and sequencing

Cells isolated from Achilles tendons of both 2-week-old and 6-week-old mice were processed separately. Cells were resuspended in PBS with 1% fetal bovine serum (FBS) at a concentration of 10,000 cells per μL. Then, 10,000 cells per sample were loaded on a Chromium Controller (10x Genomics, Pleasanton, CA, USA) for single-cell capture. Libraries were prepared using Single Cell 3′ Library & Gel Bead Kit v3 (10x Genomics) following the manufacturer's instructions. A single-cell emulsion (Gel Bead-In-EMulsions [GEMs]) was created by making barcoded cDNA unique to each individual emulsion. A recovery agent was added to break GEM, and cDNA was then amplified. A library was produced via end repair, dA-tailing, adapter ligation, post-ligation cleanup with SPRIselect, and sample index PCR. The quality and concentration of the amplified cDNA were evaluated using the Bioanalyzer (Agilent 2100) on a High Sensitivity DNA chip (Agilent, #5065-4401; Santa Clara, CA, USA). Sequencing was performed using the HiSeq X system (Illumina, San Diego, CA, USA) to generate 28/90 bp paired-end reads.

## Cell dissociation, nuclei isolation, and snRNA-seq and snATAC-seq library construction and sequencing

After obtaining cells from the mouse Achilles tendons of 2-week-old and 6-week-old mice (processed separately for each age group), nuclei were isolated following the Chromium Next GEM Single Cell Multiome ATAC+Gene Expression Reagent Bundle (10x Genomics), and all of the buffers were made according to the manufacturer's instructions. Briefly, after centrifugation, the supernatant was discarded, and the cells were resuspended in 1 mL of 1% FBS in PBS. Approximately 500,000 cells were transferred to a new tube for further lysis. To remove the supernatant, cells were centrifuged again for 5 min at 300 rcf and 4°C and resuspended in 100 mL of chilled lysis buffer. Cells were lysed for 9 min on ice, and 1 mL of wash buffer was added to stop the reaction. Then, the suspension was filtered through a 40 μm cell strainer, cells were centrifuged and resuspended in 66.2 mL of chilled Diluted Nuclei Buffer aiming to target 7000 nuclei. The final concentration of nuclei was determined, followed by transposition, GEM generation, barcoding, and library construction according to the Chromium Next GEM Single Cell Multiome ATAC+Gene Expression (10x Genomics). Libraries were sequenced with the parameters recommended by the manufacturer, using the NovaSeq 6000 (Illumina) to generate 28/90 bp paired-end reads for gene expression and 50/49 bp paired-end reads for ATAC sequencing.

## scRNA-seq and snRNA-seq analyses

Sequencing reads were processed with the Cellranger_arc (10x Genomics, v2.0.0) using the mouse reference mm10. We performed separate analyses for scRNA-seq data (collected from both 2-week and 6-week samples) and snRNA-seq data (collected from both 2-week and 6-week samples) to enable comprehensive characterization of transcriptional landscapes. From the gene expression matrix, downstream analyses were carried out using R. Quality control, filtering, data clustering and visualization, and a differential expression analysis were carried out using Seurat (*Butler et al., 2018*). For each dataset, cells with unique feature counts over 2500 or less than 200 and >5% mitochondrial counts were filtered. Then, heterotypic doublets (assuming 5% of barcodes represent doublets) were removed using DoubletFinder (*McGinnis et al., 2019*).

Unsupervised shared nearest neighbor clustering was performed with varying resolution, and the results were visualized using UMAP (*Becht et al., 2019*). DEGs among each cell cluster were

determined using the FindAllMarkers function in Seurat. The criteria of DEGs as a marker gene for the cluster was logFC >0.25, adjusted p<0.05, expression in >25% of cells. We then annotate each cluster based on DEGs (*Supplementary files 1–15*). To ensure consistency between our different sequencing approaches, we compared cell clusters identified in the scRNA-seq analysis with those from the snRNA-seq analysis (*Figure 2—figure supplement 2*).

## Pseudotime analysis

Monocle3 (*Trapnell et al., 2014*) was used to convert the snRNA-seq dataset into a cell dataset object, preprocess data, correct for batch effects, embed with dimensional reduction, and perform pseudo-temporal ordering. Cicero (*Pliner et al., 2018*) was used to generate pseudo-temporal trajectories for the snATAC-seq dataset.

## Cell-cell communication analysis

Intercellular communication networks were quantitatively inferred and analyzed using scRNA-seq data. The R package CellChat (*Jin et al., 2025*) was used to visualize the interactions among different cell groups. Two hundred twenty-nine signaling pathway families were grouped as a library to analyze cell-cell communication.

## GO analysis

The R package ClusterProfiler (*Yu et al., 2012*) was used to perform a gene enrichment analysis. The p-values were corrected by the Benjamini-Hochberg method.

## Motif enrichment analysis (ChIP seeker)

Genomic regions containing snATAC-seq peaks were annotated using ChIPSeeker (*Yu et al., 2015*) and the UCSC database on mouse (mm10).

## snATAC-seq analysis

The Cell Ranger ATAC pipeline (1.2.0) (*Satpathy et al., 2019*) was used to preprocess the data resulting from sequencing. First, Tn5 sites were mapped to the mouse reference transcriptome mm10, and duplicate reads and background cells were removed. This returned barcoded fragment files, which were loaded into Signac (*Stuart et al., 2021*) for downstream analyses using the standard Signac/Seurat pipeline. Macs2 (*Wang et al., 2008*) was run on the fragment files to call peaks using the Signac 'CallPeaks' function. Fragments were mapped to the Macs2 called peaks and assigned to cells using the Signac 'FeatureMatrix' function. Nucleosome signal strength and TSS enrichment for each cell were calculated using the Signac 'NucleosomeSignal' and 'TSSEnrichment' functions, respectively. Outliers in the QC metric categories were removed as per Signac's standard processing guidelines. Latent semantic indexing (LSI), a form of dimensional reduction, was performed using the Signac 'RunTFIDF' and 'RunSVD' functions. The UMAP hyperparameters were varied to produce consistent object shapes (using R). Once hyperparameters were chosen, the Signac/Seurat's 'RunUMAP' function was run on the LSI dimensions chosen earlier for UMAP embedding. The Signac/Seurat 'FindNeighbors' function was run using the same LSI dimensions used for UMAP to compute the nearest neighbor graph. Signac/Seurat 'FindClusters' was then run at varying resolutions. A gene activity matrix was constructed by counting ATAC peaks within the gene body and 2 kb upstream of the transcriptional start site for protein-coding genes annotated in the Ensembl database. The gene activity matrix was log-normalized prior to label transfer with the aggregated snRNA-seq Seurat object using a canonical correlation analysis (CCA). The aggregated snATAC-seq object was filtered using a 97% confidence threshold for cell-type assignment following label transfer to remove heterotypic doublets. The filtered snATAC-seq object was reprocessed with TFIDF, SVD, and batch effect correction followed by clustering and annotation based on lineage-specific gene activity. Differential chromatin accessibility between cell types was assessed with the Signac FindMarkers function for peaks detected in at least 20% of cells using a likelihood ratio test and a log-fold-change threshold of 0.25. Bonferroni-adjusted p-values were used to determine significance at an FDR<0.05.

## Integration of snRNA-seq and snATAC-seq data

snRNA-seq and snATAC-seq data were integrated using the cluster-label transfer procedure as implemented in Signac and Seurat. Each snRNA-seq sample was clustered individually, and its cluster labels

were projected onto the matching, individually clustered snATAC-seq sample or vice versa. Anchors were identified for condition-matched snRNA- and snATAC-seq samples using the FindTransfer-Anchors function, and a CCA was performed using the snRNA expression values and the snATAC-imputed gene expression values. The anchors were used to transfer cluster-label identifiers between the two data types using the TransferData function. Each cell in the query was assigned the cluster label with the highest prediction score, and label transfer was considered successful for query cells with prediction scores above 0.3.

## SCENIC analysis

To identify TFs and characterize cell states, a cis-regulatory analysis was performed using the R package SCENIC (*Aibar et al., 2017*), which infers gene regulatory networks based on co-expression and DNA motif analyses. The network activity was then analyzed for each cell to identify recurrent cellular states. In short, TFs were identified using GENIE3 and compiled into modules (regulons), which were subsequently subjected to cis-regulatory motif analysis using RcisTarget with two gene-motif rankings: 10 kb around the TSS and 500 bp upstream. Regulon activity in every cell was then scored using AUCell.

## Immunohistochemistry

Immunohistochemistry was performed as previously described (*Tsutsumi et al., 2022*). Briefly, tissue samples were fixed in 4% paraformaldehyde (PFA) overnight at 4°C and embedded in paraffin. Sections of 10 µm in thickness were deparaffinized and activated with citric acid buffer (10 mM sodium citrate, 1 mM EDTA; pH 6.0) at 80°C for 60 min in a decloaking chamber NxGen (BIOCARE Medical, CA, USA) or with 0.1% trypsin/PBS at 37°C for 30 min. After they were blocked with Blocking One solution (Nacalai Tesque, Kyoto, Japan) for 60 min, they were incubated with rabbit anti-CD55 antibody (1:100; A13918, RRID:AB_2760771, ABclonal, Woburn, MA, USA) or rat anti-CD248 (1:100; MAB7535, RRID:AB_3658829, R&D Systems, Minneapolis, MN, USA) overnight at 4°C. Following this step, they were incubated with Alexa Fluor plus 488 goat anti-rabbit antibody (1:1000; A32731, RRID:AB_2633280, Thermo Fisher Scientific, Waltham, MA, USA) or Alexa Fluor plus 488 donkey anti-rat antibody (1:1000; A21208, RRID:AB_2535794, Thermo Fisher Scientific) for 60 min. Hoechst 33342 (100 ng/mL, Thermo Fisher Scientific) was added during this incubation. The sections were then mounted with ProLong Glass Antifade Mountant (P36980, Thermo Fisher Scientific).

## Fluorescence-activated cell sorting

Cells harvested from the Achilles tendon were incubated for 60 min on ice with APC-CD55 (#131812, RRID:AB_2800632, BioLegend, San Diego, CA, USA) and a primary antibody against CD248 (#LS-B2712, RRID:AB_1664704, LSBio) using a FACS buffer (PBS containing 1% [vol/vol] FBS). After washing, cells were reacted with fluorescent-conjugated rabbit secondary antibody. Cell sorting was performed using MoFlo XDP FACS (Beckman Coulter, Brea, CA, USA). DAPI was used as the live/dead discriminator to the gate. CD55/CD248 dual positive and negative cell clusters were sorted.

## Primary cell culture

Primary TSPCs were isolated from the Achilles tendons of 2-week-old mice with collagenase digestion described above. Single-cell suspensions were cultured in the culture medium (MEMα+20% FBS+1% penicillin-streptomycin+1% [vol/vol] 100× non-essential amino acid solution [Gibco], 1% [vol/vol] 100× GlutaMAX [Gibco]). At 80–90% confluence, cells were trypsinized, centrifuged, resuspended in culture medium as passage 1 cells, and incubated in 5% $CO_2$ at 37°C, with fresh medium every 2–3 days.

## Colony formation assay

For the colony formation assay, single-cell suspensions of TSPCs (1000 cells/well) were seeded and incubated in six-well plates for 12–14 days in the growth medium and fixed with 4% PFA (Sigma-Aldrich, St. Louis, MO, USA). Then, 0.1% crystal violet solution (Wako) was used to stain the cells. Colonies of >30–50 cells were defined as a single colony unit (*Franken et al., 2006*), and the number of clusters was counted using the ImageJ package ColonyArea (*Guzmán et al., 2014*).

## RT-PCR

Total RNA was purified using TRIzol reagent (Invitrogen, Grand Island, NY, USA). Reverse transcription of mRNA was carried out using the PrimeScript RT Reagent Kit (Takara, Shiga, Japan). Quantitative PCR (qRT-PCR) was performed on cDNA with the Thunderbird SYBR mix (Toyobo, Osaka, Japan). B2m was used as a reference gene, and relative gene expression levels were calculated through the ΔCT method.

## Tenocytes, cartilage, and osteocyte differentiation

For the differentiation experiment, TSPCs were cultured in six-well plates (50,000 cells/well). Osteogenic, chondrogenic, and tenogenic differentiation were induced using a corresponding differentiation medium. The osteogenic differentiation medium contained a culture medium supplemented with 10 nM dexamethasone (Sigma-Aldrich), 5 mM β-glycerophosphate (APEXBIO), and 0.05 mM L-ascorbic acid 2-phosphate (Sigma-Aldrich). The chondrogenic differentiation medium contained a culture medium supplemented with 100 nM dexamethasone and 10 ng/mL BMP2 (Sigma-Aldrich). The tenogenic differentiation medium contained a culture medium supplemented with 10 ng/mL TGF-β1 (Peprotech, Rocky Hill, NJ, USA), 10 ng/mL GDF-5 (R&D Systems, Minneapolis, MN, USA), and 0.05 mM L-ascorbic acid 2-phosphate. After a 2-week induction period, cells were harvested for RT-PCR.

## Bio-cultured tendon (bio-tendon)

Generation of bio-tendon has been reported previously (*Kataoka et al., 2020*; *Tsutsumi et al., 2022*) Briefly, sorted cells were embedded in a 3D-culture cocktail, which is consistent with 2 mg/mL Cellmatrix (Type I-A, Nitta Gelatin Inc, Osaka, Japan), pro-survival cocktail (final concentrations: 100 nM Bcl-Xl BH4 4-23 [Merck Millipore, Burlington, MA, USA], 100 μM carbobenzoxy-valyl-alanyl-aspartyl-[O-methyl]-fluoromethylketone [Z-VAD-FMK] [Promega, Madison, WI, USA]) in culture medium. The 3D chamber was coated with 1% gelatin and incubated at 37°C and 5% $CO_2$ for 30 min. After washing with 1× PBS three times, $1.0 \times 10^6$ cells were transferred to a 3D-culture cocktail mixture and incubated at 37°C and 5% $CO_2$ for 60 min for gelation. Following gelation, tenogenic differentiation medium was added to the chamber. Following 24 hr of incubation, the chamber was placed within a cell stretching system (Shellpa Pro, Menicon Life Science, Aichi, Japan). Cyclic mechanical stretch was applied for 1 week, with gradual increase in stretch load: 1% on day 1, 2% on day 2, 3% on day 3, 4% on day 4, and 5% from days 5 to 7. The stretching cycle was programmed at 0.25 Hz for 18 hr/day, followed by resting for 6 hr/day at 37°C and 5% $CO_2$. Daily medium changes were also required.

## Decellularization by HHP and chemical treatment

The cultured bio-tendon was placed into a plastic pack filled with saline and sealed to prevent implosion and leakage during the procedure. The pack was then pressurized at 1000 MPa at 30°C for 10 min using an HHP machine (Dr. CHEF; Kobe Steel, Hyogo, Japan). After pressurization, the bio-tendon was washed thrice with 30% ethanol (EtOH) by continuous shaking for 5 min at each step. Finally, 1-ethyl-3-(3-dimethylaminopropyl) carbodiimide (EDC)/N-hydroxysuccinimide (NHS)-based cross-linking was performed by adding 70 mM EDC (Wako) and 70 mM NHS (Wako) with 30% EtOH for 24 hr at 4°C. The processed bio-tendon was incubated in 1× PBS at 4°C until the experiment.

## Electron microscopy

Bio-tendon tissues were dissected and fixed overnight in 2.5% glutaraldehyde in 0.1 M phosphate buffer (PB). For TEM, the specimens (n=3) were rinsed with 0.1 M PB, post-fixed in 1% osmium buffered with 0.1 M PB for 2 hr, and dehydrated in a graded ethanol series. The specimens were then embedded in Epon 812, sectioned into ultrathin sections (70 nm), mounted on copper grids, and double-stained with uranyl acetate and lead citrate. TEM observation was conducted on a JEM-1400Flash (JEOL, Tokyo, Japan). For SEM, the specimens (n=2) were dried in a critical-point dryer (HCP-2; Hitachi, Tokyo, Japan) with liquid $CO_2$ and coated with platinum. The specimens were subsequently observed under SEM (JSM-7900F/JED-2300; JEOL). Fiber orientation within the microscopic sections was analyzed using the OrientationJ image processing tool, a plugin for ImageJ.

## Stretch test

The mechanical properties were measured using a creep meter (RE-3305S; Yamaden, Tokyo, Japan). After measuring the initial length (mm), diameter (mm), and thickness (mm) using a micrometer, the

samples were fixed with two grips, which were pulled at a constant speed of 0.05 mm/s until failure, and the tensile strength (N) and failure strain (mm) were measured. The cross-sectional area (mm$^2$) was calculated using the initial diameter and thickness. The stiffness was manually determined from the slope in the linear region of the failure-stress curve.

The tensile strength (MPa), failure strain (%), stiffness, and Young's modulus were calculated using the following formulas:

Tensile strength (MPa)=tensile strength (N)/cross-sectional area (mm$^2$).
Failure strain (%)=failure strain (mm)/initial length (mm).
Stiffness=stress (N)/strain (mm).
Young's modulus=stiffness×initial length (mm)/cross-sectional area (mm$^2$).

## Statistical analysis

All values are presented as means ± SEM. Statistically significant differences were assessed using unpaired two-tailed Student's t-tests and one-way analysis of variance (ANOVA) with Tukey's post hoc tests. Statistical significance was set at $p < 0.05$.

## Acknowledgements

We thank all the members of the Department of Systems BioMedicine at Institute of Science Tokyo for their support. We also thank the Research Core at Institute of Science Tokyo for supporting cell sorting. This work was supported by JSPS KAKENHI (Grant Numbers JP15H02560, JP20H05696, 16H06279 [PAGS]), AMED (Grant Numbers JP21gm0810008, JP23ym0126805, JP24gm0010009, JP24jf0126010), and NIH (Grant Number R01AR080127) to HA.

## Additional information

### Funding

| Funder | Grant reference number | Author |
|---|---|---|
| National Institute of Arthritis and Musculoskeletal and Skin Diseases | R01AR080127 | Hiroshi Asahara |
| Japan Society for the Promotion of Science | JP15H02560 | Hiroshi Asahara |
| Japan Society for the Promotion of Science | JP20H05696 | Hiroshi Asahara |
| Japan Society for the Promotion of Science | 16H06279 | Hiroshi Asahara |
| Japan Agency for Medical Research and Development | JP21gm0810008 | Hiroshi Asahara |
| Japan Agency for Medical Research and Development | JP23ym0126805 | Hiroshi Asahara |
| Japan Agency for Medical Research and Development | JP24gm0010009 | Hiroshi Asahara |
| Japan Agency for Medical Research and Development | JP24jf0126010 | Hiroshi Asahara |

The funders had no role in study design, data collection and interpretation, or the decision to submit the work for publication.

## Author contributions

Hiroki Tsutsumi, Conceptualization, Investigation, Methodology, Writing - original draft; Tomoki Chiba, Conceptualization, Methodology, Writing – review and editing; Yuta Fujii, Takahide Matsushima, Tsuyoshi Kimura, Akinori Kanai, Akio Kishida, Yutaka Suzuki, Investigation; Hiroshi Asahara, Formal analysis, Supervision, Funding acquisition, Writing – review and editing

## Author ORCIDs

Tomoki Chiba ⬡ https://orcid.org/0000-0001-5472-9030
Hiroshi Asahara ⬡ https://orcid.org/0000-0002-5215-8745

## Ethics

All animal experiments were conducted in accordance with the Guidelines for the Care and Use of Laboratory Animals of Institute of Science Tokyo. The study protocol was approved by the Animal Care and Use Committee of Institute of Science Tokyo (approval number: A2024-012). All procedures conformed to the Japanese Act on Welfare and Management of Animals (Law No. 105, 1973) and the guidelines established by the Ministry of Education, Culture, Sports, Science and Technology, Japan.

Reviewer #1 (Public review): https://doi.org/10.7554/eLife.104768.3.sa1
Reviewer #2 (Public review): https://doi.org/10.7554/eLife.104768.3.sa2
Author response https://doi.org/10.7554/eLife.104768.3.sa3

# Additional files

## Supplementary files

Supplementary file 1. Diferentially expressed genes in cluster 0, related to *Figure 1A*.
Supplementary file 2. Diferentially expressed genes in cluster 1, related to *Figure 1A*.
Supplementary file 3. Diferentially expressed genes in cluster 2, related to *Figure 1A*.
Supplementary file 4. Differentially expressed genes in cluster 3, related to *Figure 1A*.
Supplementary file 5. Differentially expressed genes in cluster 4, related to *Figure 1A*.
Supplementary file 6. Differentially expressed genes in cluster 5, related to *Figure 1A*.
Supplementary file 7. Differentially expressed genes in cluster 6, related to *Figure 1A*.
Supplementary file 8. Differentially expressed genes in cluster 7, related to *Figure 1A*.
Supplementary file 9. Differentially expressed genes in cluster 8, related to *Figure 1A*.
Supplementary file 10. Differentially expressed genes in cluster 9, related to *Figure 1A*.
Supplementary file 11. Differentially expressed genes in cluster 10, related to *Figure 1A*.
Supplementary file 12. Differentially expressed genes in cluster 11, related to *Figure 1A*.
Supplementary file 13. Differentially expressed genes in cluster 12, related to *Figure 1A*.
Supplementary file 14. Differentially expressed genes in cluster 13, related to *Figure 1A*.
Supplementary file 15. Differentially expressed genes in cluster 14, related to *Figure 1A*.
Supplementary file 16. Differentially expressed genes in cluster 15, related to *Figure 1A*.
Supplementary file 17. PCR primers used in this study.
MDAR checklist

## Data availability

FASTQ data of RNA-Seq and ATAC-seq are deposited in DDBJ under accession number PRJDB18857.

The following dataset was generated:

| Author(s) | Year | Dataset title | Dataset URL | Database and Identifier |
|---|---|---|---|---|
| Chiba T, Tsutsumi H, Asahara H | 2024 | Single cell multiome analysis of mouse achilles tendon | https://ddbj.nig.ac.jp/search/entry/bioproject/PRJDB18857 | DDBJ, PRJDB18857 |

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
