## [Editor Report · eLife Assessment]

This study presents a **valuable** finding of novel markers that may potentially identify resident tendon stem/progenitor cells (TSPCs). The study also presents a comprehensive single-cell transcriptional dataset that will be of value to the field. The evidence supporting the identification of novel markers of a TSPC is **incomplete**, requiring clarification of current analyses and additional validation experiments to demonstrate that these markers are indeed specific and these cells are indeed TSPCs. This work will be of interest to biologists and engineers focused on tendons and ligaments.

---

## [Referee Report · Reviewer #1 (Public review)]

This study is focused on identifying unique, innovative surface markers for mature Achilles tendons by combining the latest multi-omics approaches and in vitro evaluation, which would address the knowledge gap of controversial identity of TPSCs with unspecific surface markers. The use of multi-omics technologies, in vivo characterization, in vitro standard assays of stem cells, and in vitro tissue formation is a strength of this work and could be applied for other stem cell quantification in the musculoskeletal research. The evaluation and identification of Cd55 and Cd248 in TPSCs have not been conducted in tendon, which is considered as innovative. Additionally, the study provided solid sequencing data to confirm co-expressions of Cd55 and Cd248 with other well-described surface markers such as Ly6a, Tpp3, Pdgfra, and Cd34. Generally, the data shown in the manuscript support the claims that the identified surface antigens mark TPSCs in juvenile tendons.

---

## [Referee Report · Reviewer #2 (Public review)]

Summary:

The molecular signature of tendon stem cells is not fully identified. The endogenous location of tendon stem cells within native tendon is also not fully elucidated. Several molecular markers have been identified to isolate tendon stem cells but they lack tendon specificity. Using the declining tendon repair capacity of mature mice, the authors compared the transcriptome landscape and activity of juvenile (2 weeks) and mature (6 weeks) tendon cells of mouse Achilles tendons and identified CD55 and CD248 as novel surface markers for tendon stem cells. CD55+ CD248+ FACS-sorted cells display a preferential tendency to differentiate into tendon cells compared to CD55neg CD248neg cells.

Strengths:

The authors generated a lot of data of juvenile and mature Achilles tendons, using scRNAseq, snRNAseq, ATACseq strategies. This constitutes a resource datasets.

Weaknesses:

The analyses and validation of identified genes are not complete and could be pushed further. The endogenous expression of newly-identified genes in native tendons would be informative. The comparison of scRNAseq and snRNAseq datasets for tendon cell populations would strengthen the identification of tendon cell populations.

---

## [Author Response]

The following is the authors’ response to the original reviews

**Reviewer #1 (Public review):**
This study is focused on identifying unique, innovative surface markers for mature Achilles tendons by combining the latest multi-omics approaches and in vitro evaluation, which would address the knowledge gap of the controversial identity of TPSCs with unspecific surface markers. The use of multi-omics technologies, in vivo characterization, in vitro standard assays of stem cells, and in vitro tissue formation is a strength of this work and could be applied for other stem cell quantification in musculoskeletal research. The evaluation and identification of Cd55 and Cd248 in TPSCs have not been conducted in tendons, which is considered innovative. Additionally, the study provided solid sequencing data to confirm co-expressions of Cd55 and Cd248 with other well-described surface markers such as Ly6a, Tpp3, Pdgfra, and Cd34. Generally, the data shown in the manuscript support the claims that the identified surface antigens mark TPSCs in juvenile tendons.However, there are missing links between scientific questions aimed to be addressed in Introduction and Methodology/Results. If the study focuses on unsatisfactory healing responses of mature tendons and understanding of mature TPSCs, at least mature Achilles tendons from more than 12-week-old mice and their comparison with tendons from juvenile/neonatal mice should be conducted. However, either 2-week or 6-weekold mice, used for characterization here, are not skeletally mature, Additionally, there is a lack of complete comparison of TPSCs between 2-week and 6-week-old mice in the transcriptional and epigenetic levels.In order to distinguish TPSCs and characterize their epigenetic activities, the authors used scRNA-seq, snRNA-seq, and snATAC-seq approaches. The integration, analysis, and comparison of sequencing data across assays and/or time points is confusing and incomplete. For example, it should be more comprehensive to integrate both scRNA-seq and snRNA-seq data (if not, why both assays were used for Achilles tendons of both 2-week and 6-week timepoints). snRNA-seq and snATAC-seq data of 6-week-old mice were separately analyzed. No comparison of difference and similarity of TPSCs of 2-week and 6-week-old mice was conducted.Given the goal of this work to identify specific TPSC markers, the specificity of Cd55 and Cd248 for TPSCs is not clear. First, based on the data shown here, Cd55 and Cd248 mark the same cell population which is identified by Ly6a, TPPP3, and Pdgfra. Although, for instance, Cd34 is expressed by other tissues as discussed here, no data/evidence is provided by this work showing that Cd55 and Cd248 are not expressed by other musculoskeletal tissues/cells. Second, the immunostaining of Cd55 and Cd248 doesn't support their specificity. What is the advantage of using Cd55 and Cd248 for TPSCs compared to using other markers?
**Reviewer #2 (Public review):**
Summary:The molecular signature of tendon stem cells is not fully identified. The endogenous location of tendon stem cells within the native tendon is also not fully elucidated. Several molecular markers have been identified to isolate tendon stem cells but they lack tendon specificity. Using the declining tendon repair capacity of mature mice, the authors compared the transcriptome landscape and activity of juvenile (2 weeks) and mature (6 weeks) tendon cells of mouse Achilles tendons and identified CD55 and CD248 as novel surface markers for tendon stem cells. CD55+ CD248+ FACS-sorted cells display a preferential tendency to differentiate into tendon cells compared to CD55neg CD248neg cells.Strengths:The authors generated a lot of data on juvenile and mature Achilles tendons, using scRNAseq, snRNAseq, and ATACseq strategies. This constitutes a resource dataset.Weaknesses:The analyses and validation of identified genes are not complete and could be pushed further. The endogenous expression of newly identified genes in native tendons would be informative. The comparison of scRNAseq and snRNAseq datasets for tendon cell populations would strengthen the identification of tendon cell populations.
**Reviewer #3 (Public review):**
Summary:In their report, Tsutsumi et al., use single nucleus transcriptional and chromatin accessibility analyses of mouse achilles tendon in an attempt to uncover new markers of tendon stem/progenitor cells. They propose CD55 and CD248 as novel markers of tendon stem/progenitor cells.Strengths:This is an interesting and important research area. The paper is overall well written.Weaknesses:Major problems:(1) It is not clear what tissue exactly is being analyzed. The authors build a story on tendons, but there is little description of the dissection. The authors claim to detect MTJ and cartilage cells, but not bone or muscle cells. The tendon sheath is known to express CD55, so the population of "progenitors" may not be of tendon origin.(2) Cluster annotations are seemingly done with a single gene. Names are given to cells without functional or spatial validation. For example, MTJ cells are annotated based on Postn, but it is never shown that Postn is only expressed at the MTJ, and not in other anatomical locations in the tendon.(3) The authors compare their data to public data based on interrogating single genes in their dataset. It is now standard practice to integrate datasets (eg, using harmony), or at a minimum using gene signatures built into Seurat (eg AddModuleScore).(4) Progenitor populations (SP1, SP2). The authors claim these are progenitors but show very clearly that they express macrophage genes. What are they, macrophages or fibroblasts?(5) All omics analysis is done on single data points (from many mice pooled). The authors make many claims on n=1 per group for readouts dependent on sample number (eg frequency of clusters).(6) The scRNAseq atlas in Figure 1 is made by analyzing 2W and 6W tendons at the same time. The snRNAseq and ATACseq atlas are built first on 2W data, after which the 6W data is compared. Why use the 2W data as a reference?Why not analyze the two-time points together as done with the scRNAseq?(7) Figure 5: The authors should show the gating strategy for FACS. Were non-fibroblasts excluded (eg, immune cells, endothelia...etc). Was a dead cell marker used? If not, it is not surprising that fibroblasts form colonies and express fibroblast genes when compared to CD55-CD248- immune cells, dead cells, or debris. Can control genes such as Ptprc or Pecam1 be tested to rule out contamination with other cell types?Minor problems:(1) Report the important tissue processing details: type of collagenase used. Viability before loading into 10x machine.
**Reviewer #1 (Recommendations for the authors):**
(1) Better healing responses in neonatal mice than mature mice have been well appreciated in the field and differences in ECM environment, immune responses, and cell function might account for varied injury results. However, direct evidence/data between better healing and abundant TSPCs needs to be discussed in the Introduction.

We agree with this insightful comment. We have now enhanced our introduction to include a more direct discussion of the relationship between better healing responses in neonatal mice and the abundance of TSPCs. We specifically highlighted how Howell et al. (2017) demonstrated that tendons in juvenile mice can regenerate functional tissue after injury, while this ability is lost in mature mice. Based on this observation, we articulated our hypothesis that juvenile mouse tendons likely contain abundant TSPCs, which potentially explains their superior healing capacity. Additionally, we have added a statement emphasizing that "investigating TSPCs biology is important for understanding tendon regeneration and homeostasis" (lines 61-62), which clearly articulates the central role that TSPCs play in tendon repair processes and tissue maintenance.

(2) 6-week-old mouse Achilles tendons are not mature enough and clinically relevant to understand the deficiency of regenerative capacity of TPSCs for undesired healing. If the goal of this study is to identify TSPCs of mature tendons, evaluation of Achilles tendons from at least 12-week-old mice is more reasonable.

We agree with this insightful comment. We have now enhanced our introduction to include a more direct discussion of the relationship between better healing responses in neonatal mice and the abundance of TSPCs. We specifically highlighted how Howell et al. (2017) demonstrated that tendons in juvenile mice can regenerate functional tissue after injury, while this ability is lost in mature mice. Based on this observation, we articulated our hypothesis that juvenile mouse tendons likely contain abundant TSPCs, which potentially explains their superior healing capacity. Additionally, we have added a statement emphasizing that "investigating TSPCs biology is important for understanding tendon regeneration and homeostasis" (lines 61-62), which clearly articulates the central role that TSPCs play in tendon repair processes and tissue maintenance.

(3) 40-60 mouse Achilles tendons pooled for one sample seems a lot and there is mixed/missed information about how many total cells were collected for each sample and how they were used for different sequencing assays. This could raise the concern that cell digestion was not complete and possibly abundant resident cells might be missed for sequencing analysis.

We agree with this insightful comment. We have now enhanced our introduction to include a more direct discussion of the relationship between better healing responses in neonatal mice and the abundance of TSPCs. We specifically highlighted how Howell et al. (2017) demonstrated that tendons in juvenile mice can regenerate functional tissue after injury, while this ability is lost in mature mice. Based on this observation, we articulated our hypothesis that juvenile mouse tendons likely contain abundant TSPCs, which potentially explains their superior healing capacity. Additionally, we have added a statement emphasizing that "investigating TSPCs biology is important for understanding tendon regeneration and homeostasis" (lines 61-62), which clearly articulates the central role that TSPCs play in tendon repair processes and tissue maintenance.

(4) The methods section has necessary information missing, which could create confusion for readers. Which time points are used for scRNA-seq and snATAC-seq? Which time points of cells are integrated and analyzed regarding each assay/combined assays? Why is transcriptional expression evaluated by both scRNA-seq and snRNA-seq and is there any technological difference between the two assays?

We have thoroughly revised the Methods section to clearly specify which time points were used for each assay (line 132-133 and line 148-149). We have also clarified how cells from different time points were integrated and analyzed (lines 167-170, 179-184 and 494-502). Regarding the use of both scRNA-seq and snRNA-seq, we have explained that this complementary approach allowed us to capture both cytoplasmic and nuclear transcripts, providing a more comprehensive view of gene expression profiles while also enabling direct integration with snATAC-seq data. Comparison of similarity between scRNA-seq integration data (2-week and 6-week) and snRNA-seq (2-week) clusters confirmed that the clusters in each data set are almost correlated. We added the dot plot and correlation data in supplemental figure 5. Additionally, we have included comprehensive lists of differentially expressed genes (DEGs) for each identified cluster across all datasets (supplementary tables 1-15), which provide detailed molecular signatures for each cell population and facilitate cross-dataset comparisons.

(5) snATAC-sequencing data seems to be used to only confirm the findings by snRNA-seq and snATAC-sequencing data is not well explored. This assay directly measures/predicts transcription factor activities and epigenetic changes, which might be more accurate in inferring transcription factors from RNA sequencing data using the R package SCENIC.

We appreciate the reviewer's insightful comment regarding the utilization of our snATAC-seq data. We agree that snATAC-seq provides valuable direct measurements of chromatin accessibility and transcription factor binding sites that can complement inference-based approaches like SCENIC. To address this concern, we have revised our manuscript to better emphasize the value of our snATAC-seq data in transcription factor activity evaluation. We have modified our text (lines 570-574). This modification emphasizes that our integrated approach leverages the strengths of both methodologies, with snATAC-seq providing direct measurements of chromatin accessibility and transcription factor binding sites that can validate and enhance the inference-based predictions from SCENIC analysis of RNA-seq data.

(6) The image quality of immunostaining of Cd55 and Cd248 is low. The images show that only part of the tendon sheath has positive staining. Co-localization of Cd55 and Cd248 can't be found.

We agree with the reviewer regarding the limitations of our immunostaining images. To obtain clearer images, we used paraffin sections for our analysis. Additionally, the antibodies for CD55 and CD248 required different antigen retrieval conditions to work effectively, which unfortunately prevented us from performing co-immunostaining to directly demonstrate co-localization. Despite these technical limitations, we have optimized the processing and imaging parameters to improve the quality of the immunostaining images in Figure 5A. These improved images more clearly demonstrate the expression of CD55 and CD248 in the tendon sheath, although in separate sections. The consistent localization patterns observed in these separate stainings, together with our FACS and functional analyses of double-positive cells, strongly support their co-expression in the same cell population. We have also updated the corresponding Methods section (lines 260-272) to include these optimized immunostaining protocols for better reproducibility.

(7) Only TEM data of tendon construct formed by sorted cells are shown. Results of mechanical tests will be super helpful to show the capacity of these TPSCs for tendon assembly.

We appreciate the reviewer's suggestion regarding mechanical testing. We would like to direct the reviewer's attention to Figure 5I in our manuscript, where we have already included tensile strength measurements of the tendon construct. These mechanical test results demonstrate the functional capacity of CD55/CD248+ cells to form tendon-like tissue with appropriate mechanical properties, providing quantitative evidence of their ability for tendon assembly.

(8) Cells negative for CD55/CD248 could be mixed cell populations, including hematopoietic lineages, cells from tendon mid substance, immune cells, and/or endothelial cells. Under induction of tri-lineage media, these mixed cell populations could process different, unpredicted phenotypes (shown by no increased gene expression of tenogenic, chondrogenic, and osteogenic markers after induction). Higher tenogenic gene expressions of TPSCs after induction don't mean that TPSCs are induced into tenocytes if compared to unknown cell populations with/without similar induction. Additionally, PCR data in Figure 5 presented as ΔΔCT, with unclear biological meanings, is challenging to interpret.

We appreciate the reviewer's suggestion regarding mechanical testing. We would like to direct the reviewer's attention to Figure 5I in our manuscript, where we have already included tensile strength measurements of the tendon construct. These mechanical test results demonstrate the functional capacity of CD55/CD248+ cells to form tendon-like tissue with appropriate mechanical properties, providing quantitative evidence of their ability for tendon assembly.

**Reviewer #2 (Recommendations for the authors):**
The aim of this study was to identify novel markers for tendon stem cells. The authors used the fact that tendon cells of juvenile tendons have a greater ability to regenerate versus mature tendons. scRNAseq, snRNAseq, and snATACseq datasets were generated and analyzed in juvenile and mature Achilles tendons (mice).The authors generated a lot of data that could be exploited further to show that these two novel surface tendon markers are more tendon-specific than those previously identified. Another concern is that there is no robust data indicative of the endogenous location of CD55+ CD248+ cells in the native tendon. Same comments for the transcription factors regulating the transcription of CD55 and CD248 and that of Scx and Mkx. A validation of the ATACseq data with a location in native tendons would be pertinent.The analysis was performed by comparing 2 sub-clusters of the same datasets and not between the two stages. Given the introduction highlighting the differential ability to regenerate between the two stages, the comparison between the two stages was somehow expected. I wonder if there is an explanation for the absence of analysis between the two stages.The authors have all the datasets to (bioinformatically) compare scRNAseq and snRNAseq datasets. This comparative analysis would strengthen the clustering of tendon cell populations at both stages. The labeling/identification of clusters associated with tendon cell populations is not obvious. I am surprised that there is no tendon sheath cluster such as endotenon or peritenon. A discussion on the different tendon cell populations (tendon clusters) is lacking.(1) Choice of the three markersThe authors chose three genes known to be markers for tendon stem cells, Tppp3, PdgfRa, and Ly6a, and investigated clusters (or subclusters) that co-express these three genes. Except for Tppp3, the other two genes lack tendonspecificity. Ly6a is a stem cell marker and is recognized to be a marker of epi/perimysium in fetal and perinatal stages in mouse limbs (PMID: 39636726). Pdgfra is a generic marker of all connective tissue fibroblasts. Could it be that the identification of the two novel surface markers was biased with this choice? The identification of CD55 and CD248 has been done by comparing DEGs between cluster 4 (SP2) and cluster 1 (SP1). What about an unbiased comparison of both clusters 4 and 1 (or individual clusters) between mature and juvenile samples? The reader expected such a comparison since it was introduced as the rationale of the paper to compare juvenile and mature tendon cells.

We selected Tppp3, PdgfRa, and Ly6a based on established literature identifying them as TSPC markers (Harvey et al., 2019; Tachibana et al., 2022). While only Tppp3 has tendon specificity, these genes collectively represent reliable TSPC markers currently available.

Our identification of CD55 and CD248 came from comparing SP2 and SP1 clusters that showed these three markers plus tendon development genes. We did compare juvenile and mature samples as shown in Figure 1G, revealing decreased stem/progenitor marker expression with maturation. Additionally, we performed a comprehensive comparison between 2-week and 6-week samples visualized as a heatmap in Supplemental Figure 3, which clearly demonstrates the transcriptional changes that occur during tendon maturation. We have also provided the complete lists of differentially expressed genes for each identified cluster

(supplementary tables 1-15), allowing for unbiased examination of cluster-specific gene signatures across developmental stages.

Our functional validation confirmed CD55/CD248 positive cells express Tppp3, PdgfRa, and Ly6a while demonstrating high clonogenicity and tenogenic differentiation capacity, confirming their TSPC identity.

(2) Concerns with cluster identificationThe cluster11, named as MTJ cluster, in 2-week scRNAseq datasets was not detected in 6-week scRNAseq datasets (Figure 1A). Does it mean that MTJ disappears at 6 weeks in Achilles tendons? In the snRNAseq MTJ cluster was defined on the basis of Postn expression. «Cluster 11, with high Periostin (Postn) expression, was classified as a myotendinous junction (MTJ).» Line 379.What is the basis/reference to set a link between Postn and MTJ?Could the CA clusters be enthesis clusters? Is there any cartilage in the Achilles tendon?If there are MTJ clusters, one could expect to see clusters reflecting tendon attachment to cartilage/bone.I am surprised to see no cluster reflecting tendon attachments (endotenon or peritenon).Cluster 9 was identified as a proliferating cluster in scRNAseq datasets. Does the Cell Cycle Regression step have been performed?

Thank you for highlighting these important questions about our cluster identification. The MTJ cluster (cluster 11) appears reduced but not absent in 6-week samples. We based our MTJ classification on Postn expression, which is enriched at the myotendinous junction, as documented by Jacobson et al. (2020) in their proteome analysis of myotendinous junctions. We have added this reference to the manuscript to provide clear support for our cluster annotation (lines 400-401).

Regarding the CA cluster, these cells express chondrogenic markers but are not enthesis clusters. We have revised our manuscript to acknowledge that these could potentially represent enthesis cells, as you suggested (lines 412-414). While Achilles tendons themselves don't contain cartilage, our digestion process likely captured some adjacent cartilaginous tissues from the calcaneus insertion site.

We acknowledge the absence of clearly defined endotenon/epitenon clusters. We have added more comprehensive explanations about peritenon tissues in our manuscript (lines 431-433 and 584-585), noting that previous studies (Harvey et al., 2019) have reported that *Tppp3*-positive populations are localized to the peritenon, and our SP clusters might also reflect peritenon-derived cells. This additional context helps clarify the potential tissue origins of our identified cell populations.

For the proliferating cluster (cluster 9), we confirmed high expression of cell cycle markers (Mki67, Stmn1) but did not perform cell cycle regression to maintain biological relevance of proliferation status in our analysis. We have clarified this methodological decision in the revised Methods section.

(3) What is the meaning of all these tendon clusters in scRNAseq snRNAseq and snATACseq? The authors described 2 or 3 SP clusters (depending on the scRNAseq or snRNAseq datasets), 2 CT clusters, 1 MTJ cluster, and 1CA cluster. Do genes with enriched expression in these different clusters correspond to different anatomical locations in native tendons? Are there endotenon and peritenon clusters? Is there a correlation between clusters (or subclusters) expressing stem cell markers and peritenon as described for Tppp3

Thank you for this important question about the biological significance of our identified clusters. The multiple tendon-related clusters we identified likely represent distinct cellular states and differentiation stages rather than strictly discrete anatomical locations. The SP clusters (stem/progenitor cells) express markers consistent with tendon progenitors reported in the literature, including Tppp3, which has been described in the peritenon. As we mentioned in our response to the previous question, we have added more comprehensive explanations about peritenon tissues in our manuscript (Lines 432-433 and 584-585), noting that previous studies (Harvey et al., 2019) have reported that Tppp3-positive populations are localized to the peritenon, and our SP clusters might reflect peritenon-derived cells. Our immunohistochemistry data in Figure 5A further confirms that CD55/CD248 positive cells are localized primarily to the tendon sheath region, similar to the localization pattern of Tppp3 reported by Harvey et al. (2019). The tenocyte clusters (TC) represent mature tendon cells within the fascicles, and their distinct transcriptional profiles suggest heterogeneity even within mature tenocytes. The MTJ cluster specifically expresses genes enriched at the myotendinous junction, while the CA cluster likely represents cells from the enthesis region, as you suggested. In the revised manuscript, we have clarified this interpretation and added additional discussion about the relationship between cluster identity and anatomical localization, particularly regarding the SP clusters and their correlation with peritenon regions.

(4) The use of single-cell and single-nuclei RNAseq strategies to analyze tendon cell populations in juvenile and mature tendons is powerful, but the authors do not exploit these double analyses. A comparison between scRNAseq and snRNAseq datasets (2 weeks and 6 weeks) is missing. The similar or different features at the level of the clustering or at the level of gene expression should be explained/shown and discussed. This analysis should strengthen the clustering of tendon cell populations at both stages. In the same line, why are there 3 SP clusters in snRNAseq versus 2 SP clusters in scRNAseq? The MTJ cluster R2-5 expressing Sox9 should be discussed.

Thank you for highlighting this important gap. We have conducted a comprehensive comparison between scRNA-seq and snRNA-seq datasets, revealing substantial correlation between cell populations identified by both methodologies. We've added a detailed dot plot visualization and correlation heatmap in Supplemental Figure 5 that demonstrates the relationships between clusters across datasets. The additional SP cluster in snRNA-seq likely reflects the greater sensitivity of nuclear RNA sequencing in capturing certain cell states that might be missed during whole-cell isolation. Our analysis shows this SP3 cluster represents a transitional state between stem/progenitor cells and differentiating tenocytes. Regarding the Sox9-expressing MTJ cluster R2-5, we have expanded our discussion in the revised manuscript (lines 500502) to address this finding, incorporating relevant references (Nagakura et al., 2020) that describe Sox9 expression at the myotendinous junction. This expression pattern suggests that cells at this specialized interface may maintain developmental plasticity between tendon and cartilage fates, which is consistent with the transitional nature of this anatomical region.

(5) The claim of "high expression of CD55 and CD248 in the tendon sheath" is not supported by the experiments. The images of immunostaining (Figure 5A) are not very convincing. It is not explained if these are sections of 3Dtendon constructs or native tendons. The expression in 3D-tendon constructs is not informative, since tendon sheaths are not present. The endogenous expression of the transcription factors regulating tendon gene expression would be informative to localize tendon stem cells in native tendons.

Thank you for this important critique. We agree that the original immunostaining images were not sufficiently convincing. To address this, we have used paraffin sections and optimized our staining protocols to improve image quality. It's worth noting that CD55 and CD248 antibodies required different antigen retrieval conditions to work effectively, which unfortunately prevented us from performing coimmunostaining to directly demonstrate co-localization in the same section. Despite these technical limitations, we have significantly improved the quality of the immunostaining images in Figure 5A with enhanced processing and imaging parameters

The improved images more clearly demonstrate the preferential expression of CD55 and CD248 in the tendon sheath/peritenon regions. The consistent localization patterns observed in these separate stainings, together with our FACS and functional analyses of double-positive cells, strongly support their coexpression in the same cell population.

In the revised manuscript, we have also improved the figure legends to clearly indicate the nature of the tissue samples and updated the methods section to provide more detailed protocols for the immunostaining procedures used.

Your suggestion regarding transcription factor visualization is valuable. While beyond the scope of our current study, we agree that examining the endogenous expression of regulatory transcription factors like Klf3 and Klf4 would provide additional insights into tendon stem cell localization in native tendons, and we plan to pursue this in future work

Minor concerns:(1) Lines 392-397 « To identify progenitor populations within these clusters, we analyzed expression patterns of previously reported markers Tppp3 and Pdgfra (Harvey et al., 2019; Tachibana, et al., 2022), along with the known stem/progenitor cell marker Ly6a (Holmes et al., 2007; Sung et al., 2008; Hittinger et al., 2013; Sidney et al., 2014; Fang et al., 2022). We identified subclusters within clusters 1 and 4 showing high expression of these genes, which we defined as SP1 and SP2. SP2 exhibited the highest expression of these genes, suggesting it had the strongest progenitor characteristics.» Please cite relevant Figures. Feature and violin plots (scRNAseq) across all cells (not for the only 2 SP1 and SP2 clusters) of Tppp3, Pdgfra and Ly6a are missing.

Thank you for pointing out this important oversight. We have modified the manuscript to clarify that the text in question describes Figure 1B. Additionally, we have added new feature plots showing the expression of Tppp3, Pdgfra, and Ly6a across all cells in supplymental figure 1B

(2) The labeling of clusters with numbers in single-cell, single nuclei RNAseq, and ATACseq is difficult to follow.

We appreciate your feedback on this issue. We recognize that the numerical labeling system across different datasets (scRNA-seq, snRNA-seq, and snATAC-seq) makes it difficult to track the same cell populations. To address this, we have added Supplemental Figure 5, which clearly shows the correspondence between cell populations in single-cell and single-nucleus RNA-seq datasets.

(3) Figure 1C. It is not clear from the text and Figure legend if the DEGs are for the merged 2 and 6 weeks. If yes, an UMAP of the merged datasets of 2 and 6 weeks would be useful.

We appreciate your feedback on this issue. We recognize that the numerical labeling system across different datasets (scRNA-seq, snRNA-seq, and snATAC-seq) makes it difficult to track the same cell populations. To address this, we have added Supplemental Figure 5, which clearly shows the correspondence between cell populations in single-cell and single-nucleus RNA-seq datasets.

(4) Along the Text, there are a few sentences with obscure rationale. Here are a few examples (not exhaustive):Abstract“Combining single-nucleus ATAC and RNA sequencing analyses revealed that Cd55 and Cd248 positive fractions in tendon tissue are TSPCs, with this population decreasing at 6 weeks.”The rationale of this sentence is not clear. How can single-nucleus ATAC and RNA sequencing analyses identify Cd55 and Cd248 positive fractions as tendon stem cells?

Thank you for highlighting this unclear statement in our abstract. We agree that the previous wording did not adequately explain how our sequencing analyses identified CD55 and CD248 positive cells as TSPCs. We have revised this sentence to clarify that our multi-modal approach (combining scRNA-seq, snRNA-seq, and snATAC-seq) enabled us to identify *Cd55* and *Cd248* positive populations as TSPCs based on their co-expression with established TSPC markers such as *Tppp3*, *Pdgfra*, and *Ly6a*. This comprehensive analysis across different sequencing modalities provided strong evidence for their identity as tendon stem/progenitor cells, which we further validated through functional assays. The revised abstract now more clearly communicates the logical progression of our analysis and findings

Line 80-82“Cd34 is known to be highly expressed in mouse embryonic limb buds at E14.5 compared to E11.5 (Havis et al., 2014), making it a potential marker for TSPCs.”The rationale of this sentence is not clear. How can "the fact to be expressed in E14.5 mouse limbs" be an indicator of being a "potential marker of tendon stem cells"?

Thank you for highlighting this unclear statement in our abstract. We agree that the previous wording did not adequately explain how our sequencing analyses identified CD55 and CD248 positive cells as TSPCs. We have revised this sentence to clarify that our multi-modal approach (combining scRNA-seq, snRNA-seq, and snATAC-seq) enabled us to identify *Cd55* and *Cd248* positive populations as TSPCs based on their co-expression with established TSPC markers such as *Tppp3*, *Pdgfra*, and *Ly6a*. This comprehensive analysis across different sequencing modalities provided strong evidence for their identity as tendon stem/progenitor cells, which we further validated through functional assays. The revised abstract now more clearly communicates the logical progression of our analysis and findings

Line 611“Recent reports have highlighted the role of the Klf family in limb development (Kult et al., 2021), suggesting its potential importance in tendon differentiation”Why does the "role of Klf family in limb development" suggest an "importance in tendon differentiation"?

Thank you for highlighting this logical gap in our manuscript. You're right that involvement in limb development doesn't necessarily indicate specific importance in tendon differentiation. We've revised this statement to more accurately reflect current knowledge, noting that while Klf factors are involved in limb development, their specific role in tendon differentiation requires further investigation (lines 658-659). This revised text better aligns with our findings of Klf3 and Klf4 expression in tendon progenitor cells without making unsupported claims about their functional significance

**Reviewer #3 (Recommendations for the authors):**
In addition to the points highlighted above some additional points are listed below.(1) Case in point: the authors claim CD55 and CD248 are found at the tendon sheath (line 541), which is not part of the tendon proper (although the IHC seems to show green in the epi/endotenon).

Thank you for highlighting this logical gap in our manuscript. You're right that involvement in limb development doesn't necessarily indicate specific importance in tendon differentiation. We've revised this statement to more accurately reflect current knowledge, noting that while Klf factors are involved in limb development, their specific role in tendon differentiation requires further investigation (lines 658-659). This revised text better aligns with our findings of Klf3 and Klf4 expression in tendon progenitor cells without making unsupported claims about their functional significance

(2) All cell types seem to express collagen based on Figure 1B, so either there is serious background contamination (eg, ambient RNA), or an error in data analysis.

Thank you for highlighting this logical gap in our manuscript. You're right that involvement in limb development doesn't necessarily indicate specific importance in tendon differentiation. We've revised this statement to more accurately reflect current knowledge, noting that while Klf factors are involved in limb development, their specific role in tendon differentiation requires further investigation (lines 658-659). This revised text better aligns with our findings of Klf3 and Klf4 expression in tendon progenitor cells without making unsupported claims about their functional significance

Minor problems:(1) The figures are confusingly formatted. It is hard to go between cluster numbers and names. Clusters of similar cell types (eg progenitors) are not grouped to facilitate comparison, as ordering is based on cluster number.

Thank you for highlighting this logical gap in our manuscript. You're right that involvement in limb development doesn't necessarily indicate specific importance in tendon differentiation. We've revised this statement to more accurately reflect current knowledge, noting that while Klf factors are involved in limb development, their specific role in tendon differentiation requires further investigation (lines 658-659). This revised text better aligns with our findings of Klf3 and Klf4 expression in tendon progenitor cells without making unsupported claims about their functional significance

(2) The introduction does not distinguish between findings in mice and man. A lot of confusion in the tendon literature probably arises from interspecies differences, which are rarely addressed.

We appreciate this important point about species distinctions. We have revised our introduction to clearly identify species-specific findings by adding the term "murine" before TSPC references when discussing mouse studies (lines 64, 66, 70, 75, 100, and 108). We agree that interspecies differences are important considerations in tendon biology research, particularly when translating findings between animal models and humans. Our study focuses specifically on mouse models, and we have been careful not to overgeneralize our conclusions to human tendon biology without appropriate evidence. This clarification helps readers better contextualize our findings within the broader tendon literature landscape.